# SA-ResGS: Self-Augmented Residual 3D Gaussian Splatting for Next Best View Selection

## Abstract

We propose Self-Augmented Residual 3D Gaussian Splatting (SA-ResGS), a novel framework for stabilizing uncertainty quantification and enhancing uncertainty-aware supervision in next-best-view (NBV) selection for active scene reconstruction. SA-ResGS improves both the reliability of uncertainty estimates and their effectiveness for supervision by generating Self-Augmented point clouds (SA-Points) via triangulation between a training view and a rasterized extrapolated view, enabling efficient scene coverage estimation. While improving scene coverage through physically guided view selection, SA-ResGS also addresses the challenge of under-supervised Gaussians, exacerbated by sparse and wide-baseline views, by introducing the first residual learning strategy tailored for 3D Gaussian Splatting. This targeted supervision enhances gradient flow in high-uncertainty Gaussians by combining uncertainty-driven filtering with dropout- and hard-negative-mining-inspired sampling. Our contributions are threefold: (1) a physically grounded view selection strategy that promotes efficient and uniform scene coverage; (2) an uncertainty-aware residual supervision scheme that amplifies learning signals for weakly contributing Gaussians, improving training stability and uncertainty estimation across scenes with diverse camera distributions; (3) an implicit unbiasing of uncertainty quantification as a consequence of constrained view selection and residual supervision, which together mitigate conflicting effects of wide-baseline exploration and sparse-view ambiguity in NBV planning. Experiments on active view selection demonstrate that SA-ResGS outperforms state-of-the-art baselines in both reconstruction quality and view selection robustness.

## 1 Introduction

Recent advances in neural rendering—particularly Neural Radiance Fields (NeRFs) (Mildenhall et al., 2020) and 3D Gaussian Splatting (3DGS) (Kerbl et al., 2023)—have significantly advanced photorealistic scene reconstruction (Yu et al., 2024; Niedermayr et al., 2024; Kulhanek et al., 2024), enabling high-fidelity, real-time applications across diverse environments. Beyond static scene capture, these methods have spurred broader interest in tackling complex challenges, such as active view selection (Wang et al., 2023; Xiao et al., 2024; Chen et al., 2024), uncertainty quantification for next-best-view (NBV) selection (Wen et al., 2024). While pre-captured, dense-view training methods can achieve impressive reconstruction quality, in-situ (active) reconstruction—where views are selected and added progressively—remains challenging due to artifacts caused by shape-radiance ambiguity and further exacerbated by limited training views and dynamics of view addition strategy. Despite the inherent difficulty of estimating reliable uncertainty under such conditions, recent post-hoc approaches—such as Laplacian approximation-based, model-agnostic methods (Wen et al., 2024; Goli et al., 2024)—have shown promise by providing uncertainty signals without altering the rendering pipeline. However, several critical challenges remain unaddressed:

- **Disregarded physical constraints:** Computational uncertainty is often misaligned with physical plausibility of the reconstructed geometry.

- **Underutilized supervision:** Existing methods rarely convert uncertainty cues into actionable learning signals, leaving weakly contributing Gaussians under-supervised throughout training.

- **Performance dependency:** The reliability of uncertainty estimation remains tightly coupled with training dynamics, particularly in the early stages when scene coverage is incomplete.

In response to these challenges, we propose SA-ResGS, a Self-Augmented Residual 3D Gaussian Splatting framework that stabilizes uncertainty quantification and enhances uncertainty-aware supervision in next-best-view selection for progressive scene reconstruction, as shown in Fig. 1. SA-ResGS strategically decouples view selection from strong dependence on uncertainty estimates driven by internal learning dynamics, promoting more robust and geometry-aware surface coverage. Concretely, we physically prefilter a subset of candidate views based on their geometric dissimilarity and then apply uncertainty-based scoring within this subset—effectively implementing a physically grounded, uncertainty-informed selection strategy. To guide this process, we construct SA-Points by triangulating dense correspondences between a given training view and its rasterized extrapolated views at each view-selection step, after a fixed number of initial views have been trained. These SA-Points are then encoded using a hash-based scene representation, enabling efficient similarity measurement between candidate views and pre-selected views. We select the most dissimilar candidate views in the encoded space, encouraging coverage of previously unseen regions.

While the physically grounded and uncertainty-informed selection strategy enhances overall scene coverage, it inadvertently increases sparsity in multiview overlap—since more dissimilar views are less likely to observe shared regions, thereby weakening multi-view geometric constraints. To counterbalance this contradiction without sacrificing the benefits of diverse view selection, we introduce a residual supervision mechanism that reinforces training using ground truth RGB images. The proposed SA-ResGS—particularly its residual supervision module—is fully compatible with conventional 3DGS pipelines that rely on point-cloud-based rasterization and gradient-based optimization. Beyond the original supervision, SA-ResGS additionally rasterizes color images using a selected Gaussian subset: a small fraction of the most uncertain Gaussians is combined with a majority subset (e.g., 90%) of the original visible Gaussians. This strategy mirrors the principles of residual learning and draws conceptual inspiration from Dropout (Park et al., 2025; Srivastava et al., 2014) and Hard Negative Mining (Xuan et al., 2020; Jang et al., 2019), by amplifying supervision for Gaussians that typically receive weak gradients during backpropagation—similar to the role of skip connections in ResNet (He et al., 2016) in mitigating vanishing gradients. Our method thus provides additional learning signals specifically targeted at under-optimized Gaussians which are often overlooked due to their low contribution in 3DGS's rendering mechanism, receive repeated 2D photometric supervision, resulting in more stable optimization and improved reconstruction in sparse or ambiguous regions.

The main contributions of the SA-ResGS are threefold:

- **Physically grounded view selection:** We propose a geometry-aware strategy using Self-Augmented Points (SA-Points) to guide next-best-view selection, enforcing physical plausibility and promoting more balanced, coverage-oriented exploration.
- **Residual learning for 3DGS:** We introduce the first residual supervision framework specifically designed for 3D Gaussian Splatting, addressing the vanishing gradient problem by reinforcing weakly supervised Gaussians and improving both optimization stability and reconstruction quality.
- **Unbiased uncertainty quantification:** By jointly improving view distribution and supervising under-optimized Gaussians, SA-ResGS mitigates geometric sparsity and density bias, leading to fairer and more reliable uncertainty estimates throughout training.

## 2 RELATED WORKS

**Next-best-view selection.** NBV selection originated from the robotics community as a strategy to efficiently guide in-situ scene capture, where the goal was to incrementally select viewpoints that maximally reduce reconstruction ambiguity with minimal sensor movement (Connolly, 1985; Scott et al., 2003; Delmerico et al., 2018). Classical NBV methods typically followed rule-based paradigms, selecting views based on geometric coverage (Dunn and Frahm, 2009; Guédon et al., 2022; Guédon et al., 2023), viewpoint entropy (Vázquez et al., 2001), or visibility heuristics (Bircher et al., 2016; Sun et al., 2021). As NBV became increasingly relevant to 3D computer vision, especially under sparse-view constraints, learning-based approaches emerged to model scene-specific view policies via reinforcement learning or active learning frameworks (Wang et al., 2024). While these data-driven methods demonstrate improved adaptability over hand-crafted rules, they often rely on task-specific reward definitions and struggle to generalize across scene types.

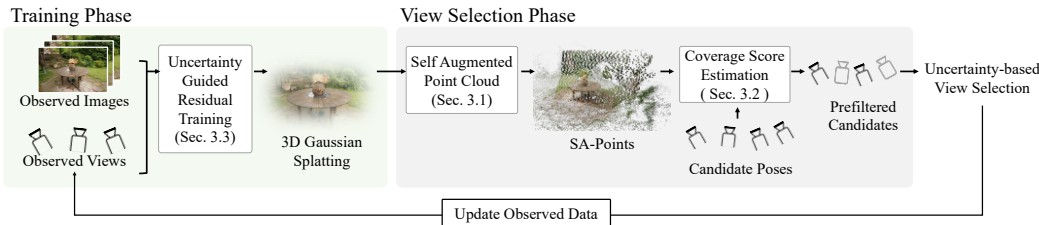

Figure 1: **Overview of SA-ResGS.** The framework alternates between view selection and training. At each NBV step, Self-Augmented Points are generated via triangulation from dense correspondences between a training view and its extrapolated render, enabling surface-aware coverage estimation (Sec.4.1). Candidate views are first physically filtered using hash-encoded feature dissimilarity, then ranked by uncertainty quantification scores for final selection (Sec.4.2). During training, residual supervision (Sec. 4.4) combines full and uncertainty-intensified renders to reinforce gradients toward weakly contributing Gaussians, improving training stability and reconstruction quality under sparse-view conditions.

More recent work explore representation-aware NBV strategies based on NeRF and 3DGS. Approaches such as NARUTO Feng et al. (2024), which learns an grid-based uncertainty field, and ActiveGAMER Chen et al. (2025), which derives uncertainty from 3DGS visibility, show how representation-specific cues can guide informative view selection. Building on this direction, information-theoretic models such as FisherRF Wen et al. (2024) offer a principled formulation for uncertainty-based NBV in neural fields. In our work, we build on this line (Wen et al., 2024; Goli et al., 2024; Hanson et al., 2024; Wilson et al., 2025) by integrating physically grounded geometry priors to further stabilize early-stage view planning, particularly when minimal visual input available.

**Uncertainty quantification for 3DGS and neural rendering.** Uncertainty quantification plays a pivotal role in active reconstruction, particularly for guiding view selection and supervision. In 3D Gaussian Splatting (3DGS), however, the high dynamic nature of primitive splitting and sensitivity to initialization often leads to unstable training, degrading the reliability of intermediate uncertainty signals. Earlier methods based on variational inference (Shen et al., 2021; Lee et al., 2025; Shen et al., 2022; Lyu et al., 2024) and ensemble-based estimates (Sünderhauf et al., 2023) enable stochastic or distributional reasoning, but they require costly model retraining or parallel inference and are typically incompatible with standard rendering pipelines. Recent post-hoc approaches (Wilson et al., 2025; Hanson et al., 2024) such as FisherRF (Wen et al., 2024) and BayesRays (Goli et al., 2024) estimate uncertainty using Laplacian approximations without altering model structure, and have demonstrated promising results on NeRF and 3DGS variants. Complementary to geometry-based methods, image-level approach (Wang et al., 2025b) leverage perceptual quality on current rendering result as a proxy for uncertainty. However, these methods remain strongly coupled with the density of underlying Gaussians—leading to biased uncertainty estimates in early training stages when geometry is sparse or unevenly distributed, often misinterpreting under-observed regions as confident. This overlooked bias limits the reliability of NBV guidance when it's most needed. To mitigate limitation, we introduce residual learning with the help of using physically grounded view selection, enabling robust and a little loose-coupled uncertainty estimation in an early-stage of view selection while emphasizing the effect of targeting high-uncertainty Gaussian focused supervision.

**Residual supervision in 3DGS.** While accurate uncertainty estimation helps localize regions requiring stronger supervision, it alone does not guarantee that gradients effectively reach under-optimized Gaussians in the 3DGS pipeline. Residual learning, as popularized by ResNet (He et al., 2016), has proven effective in mitigating vanishing gradients and improving training stability through skip connections and additive refinement, yet it remains underexplored in the context of 3D Gaussian Splatting. Existing 3DGS methods mainly rely on direct photometric losses (Kerbl et al., 2023) or external depth priors (Li et al., 2024; Xu et al., 2024), which often fail to sufficiently supervise Gaussians with low opacity or minimal rendering contributions. Recent studies such as pixelSplat (Charatan et al., 2024), PAPR (Zhang et al., 2023), and PAPR-in-Motion (Peng et al., 2024) explicitly discuss the vanishing gradient issue and propose solutions including differentiable parameterization of Gaussians, proximity attention-based differentiable renderer, adaptive updates, and activation tuning. Despite

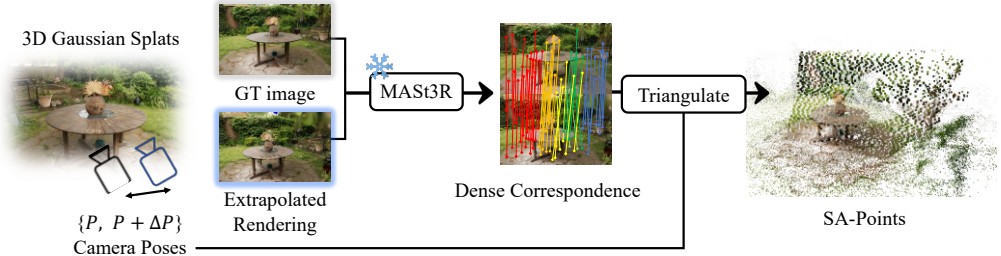

Figure 2: **SA-Points Generation.** An extrapolated image is rendered from a perturbed camera pose. Dense correspondences with the reference image are predicted using MASt3R, and triangulated to produce SA-Points, which are filtered by reprojection error for reliable surface geometry.

various strategies proposed to mitigate the vanishing gradient problem, prior approaches lack an explicit mechanism for correcting weakly supervised Gaussians, which remain largely unresolved due to insufficient gradient signals. Although dropout-based approaches (Park et al., 2025) help increase gradient diversity, they operate stochastically and do not target supervision to the most uncertain or least updated Gaussians. Our method addresses these limitations by introducing the first residual supervision strategy for 3DGS, applying uncertainty-guided rendering to intentionally amplify gradients for under-supervised Gaussians—without altering underlying rasterization process.

## 3 SELF-AUGMENTED RESIDUAL 3D GAUSSIAN SPLATTING

The proposed SA-ResGS framework is illustrated in Fig. 1. SA-ResGS builds upon the state-of-the-art next-best-view selection method, FisherRF (Wen et al., 2024), extending it with SA-Points to support two core ideas: (1) guiding physically grounded view selection with reduced reliance on uncertainty estimation, and (2) applying residual supervision to uncertain Gaussians, mitigating the vanishing gradient problem, wherein weakly contributing Gaussians—those with minimal impact on rasterized pixel values—receive insufficient gradients during backpropagation. The physically grounded view selection is enabled by a geometrically encoded surface representation, constructed using SA-Points derived from a single training view. To construct SA-Points, we employ the 3D vision foundational model, MASt3R (Leroy et al., 2024), to predict dense correspondences between a given training image and a rasterized extrapolated view rendered from a nearby camera pose. The resulting 2D correspondences are then triangulated to produce 3D SA-Points.

### 3.1 SELF-AUGMENTED POINTS GENERATION

Given a reference image $I_r$ with camera pose $\mathbf{T}_r = [\mathbf{R}_r \mid \mathbf{t}_r]$, we render an extrapolated image $I_e$ from a perturbed pose $\mathbf{T}_e = [\mathbf{R}_r \mid \mathbf{t}_r + \Delta \mathbf{t}]$ using 3D Gaussian Splatting (Kerbl et al., 2023). Dense correspondences $\{(\mathbf{p}_r^i, \mathbf{p}_e^i)\}$ between $I$ and $I_e$ are predicted using the pretrained MASt3R model (Leroy et al., 2024), which is robust to moderate viewpoint changes and capable of producing contextually meaningful matches even in the presence of minor geometric distortions. Each SA-Point $\mathbf{X}^i$ is triangulated from a 2D correspondence pair using the projection matrices $\mathbf{P}_r$ and $\mathbf{P}_e$ derived from COLMAP, including intrinsic matrices over each extrinsic pose $\mathbf{T}$. However, because triangulation is performed repeatedly during training—while the model is still fitting to a sparse and incomplete geometry—rasterized extrapolated images may occasionally contain rendering noise due to inaccurately placed Gaussians. To ensure reliable geometry while fully leveraging the generalization capability of MASt3R, we apply reprojection error-based filtering:

$$\varepsilon^i = \frac{1}{2} \left( \left\| \mathbf{p}_r^i - \pi(\mathbf{P}_r \mathbf{X}^i) \right\| + \left\| \mathbf{p}_e^i - \pi(\mathbf{P}_e \mathbf{X}^i) \right\| \right), \quad \text{retain if } \varepsilon^i < \tau. \tag{1}$$

This filtering step discards geometrically inconsistent points while preserving accurate SA-Points from dense, context-aware matches—even when the extrapolated image is noisier than the original training view. Compared to prior methods such as CoMapGS (Jang and Pérez-Pellitero, 2025) or MP-SfM (Pataki et al., 2025), our triangulation pipeline produces scale-consistent, surface-aware geometry from a single image by leveraging extrapolated viewpoints rather than requiring multiview input or monocular depth estimates. The overall steps are visualized in Fig. 2.

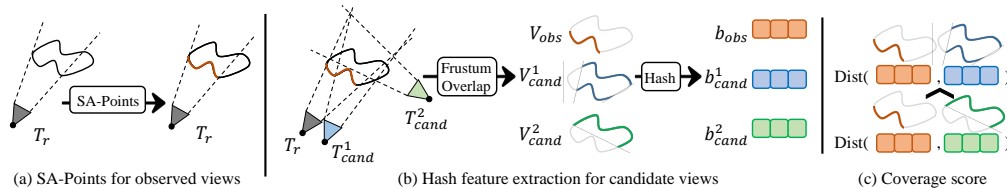

(a) SA-Points for observed views    (b) Hash feature extraction for candidate views    (c) Coverage score

Figure 3: **Physically grounded candidate view selection via surface coverage.** (a) SA-Points from training views define observed voxels $\mathcal{V}_{\text{obs}}$. (b) Each candidate view generates a binary hash-encoded feature $\mathbf{b}$, via frustum-based visibility estimation. (c) Normalized Hamming distance between hash-encoded features quantifies coverage dissimilarity, enabling efficient selection of geometrically complementary views without rendered images or uncertainty scores.

### 3.2 PHYSICALLY GROUNDED VIEW SELECTION ALGORITHM

We present our physically grounded view selection algorithm for next-best-view (NBV) selection, illustrated in Fig. 3. As discussed in Sec. 2, NBV selection in 3D Gaussian Splatting (3DGS) is particularly challenging due to the tight coupling between uncertainty estimation and the quality of re-constructed geometry—both of which are highly sensitive to the sparsity and distribution of Gaussian splats. Under sparse-view settings, where reconstruction begins with as few as four images and new views are incrementally added every 100 training iterations, uncertainty-based NBV strategies often become unreliable. This is because uncertainty signals are inherently biased or unstable when the geometry is incomplete or under-constrained. To address this, we introduce a surface-aware guidance mechanism based on SA-Points, enabling view selection to operate independently of the computed uncertainty quantification. By decoupling view selection from the internal training dynamics of 3DGS, our method provides more stable and physically meaningful candidate views during the early reconstruction phase—even before the model has accumulated sufficient confidence to produce reliable uncertainty maps.

We begin by discretizing the 3D scene into a voxel grid $\mathcal{V} = \{v_k\}_{k=1}^{K}$, where each voxel represents a unit volume in the scene. The bounding volume of $\mathcal{V}$ is defined by the sparse point cloud obtained from structure-from-motion (SfM). A voxel $v_k \in \mathcal{V}$ is marked as observed if it intersects with any SA-Point $\mathbf{X}^i$ (Sec. 4.1), forming the subset $\mathcal{V}_{\text{obs}} \subset \mathcal{V}$. To account for potential localization errors and promote coverage continuity, we dilate each occupied voxel using a 3D kernel $\mathcal{K}_r$ of radius $r$:

$$\tilde{\mathcal{V}}_{\text{obs}} = \bigcup_{v_k \in \mathcal{V}_{\text{obs}}} \mathcal{K}_r(v_k), \tag{2}$$

where $\tilde{\mathcal{V}}_{\text{obs}}$ denotes the dilated observed region for the current set of training views.

For each candidate view $j$, we compute a frustum $\mathcal{F}_j \subset \mathcal{V}$, derived from camera intrinsics (field of view) and near/far planes estimated from the SfM point distribution. A voxel is considered potentially visible from view $j$ if its center lies within the frustum:

$$\mathcal{V}_{\text{cand}}^{(j)} = \{v_k \in \mathcal{V} \mid v_k \in \mathcal{F}_j\}. \tag{3}$$

To estimate the geometric dissimilarity between the current coverage and a candidate view (see Fig. 3), we compute the normalized Hamming distance:

$$d_j = \frac{1}{K} \left\| \mathbf{b}_{\text{obs}} \oplus \mathbf{b}_{\text{cand}}^{(j)} \right\|_1, \tag{4}$$

where $\oplus$ denotes the element-wise XOR operation between binary vectors , and $\| \cdot \|_1$ is the $\ell_1$ norm (i.e., the number of differing entries). Here, $\mathbf{b}$ denotes a binary occupancy vector obtained by mapping voxel coordinates through a fixed random hashing function, following the spatial hashing strategy introduced in Instant-NGP (Müller et al., 2022). The resulting value $d_j \in [0, 1]$ measures the proportion of voxels with inconsistent occupancy status between the currently observed volume and the candidate view. Candidate views are then ranked in descending order of their normalized Hamming distances $d_j$, and the top $N\%$ (e.g., $N = 20$) are retained to form the physically filtered candidate set $\mathcal{C}'$.

Figure 4: **Residual supervision in 3DGS.** (a) At each iteration $(t_1, t_2)$, $\mathcal{G}_{\text{sup}}$ combines random and top-uncertain Gaussians; (b) residual full supervision in 3DGS mimics ResNet-style skip connections.

We apply uncertainty quantification only within $\mathcal{C}'$, finalizing the view selection with finer-level scoring. This two-stage pipeline follows a coarse-to-fine strategy: it first expands the observed surface area using explicit geometric cues from SA-Points, then refines the choice using uncertainty-aware reasoning. By restricting uncertainty estimation to a smaller candidate pool, this approach improves computational efficiency while maintaining scene-aware diversity in the selected views. This strategy enables balanced, physically grounded exploration of unobserved regions.

### 3.3 UNCERTAINTY-GUIDED RESIDUAL LEARNING IN 3DGS

We propose the first residual learning for 3DGS to address the vanishing gradient issue affecting weakly contributing Gaussians, as shown in Fig. 4. These Gaussians often receive insufficient supervision during training due to their limited impact on rasterized pixels—particularly in sparse or ambiguous regions. While ResNet (He et al., 2016) mitigates similar issues via skip connections, such mechanisms are infeasible in 3DGS due to the dynamic and view-dependent nature of Gaussian properties. Instead, we propose a rasterizer-agnostic strategy that enhances gradient flow by generating auxiliary renders that emphasize high-uncertainty Gaussians. These images are supervised with ground-truth RGB images, forming the basis for a residual supervision scheme detailed below.

**Residual supervision.** To reinforce under-supervised Gaussians, we introduce a residual supervision scheme that leverages two rasterized images from the same training viewpoint: one using the full set of Gaussians $\mathcal{G}$ and another from a guided subset $\mathcal{G}_{\text{sup}}$, as shown in Fig. 4(a). We define this subset as:

$$\mathcal{G}_{\text{sup}} = \mathcal{G}_{\text{rand}} \cup \mathcal{G}_{\text{uncertain}}, \tag{5}$$

where $\mathcal{G}_{\text{rand}}$ is a random sample comprising $\alpha\%$ of $\mathcal{G}$ (e.g., $\alpha=90$), and $\mathcal{G}_{\text{uncertain}}$ contains the top-$\beta$ most uncertain Gaussians (e.g., $\beta=10$). To estimate uncertainty, we analyze two per-Gaussian attributes: opacity and scale. Gaussians with low opacity contribute minimally to alpha blending during rasterization, while those with large scale blur across pixels and tend to dominate ambiguous or low-texture regions. This rank identifies Gaussians that are both visually suppressed and spatially diffuse—making them key targets for correction. The combination $\mathcal{G}_{\text{sup}}$ ensures that $\mathcal{G}_{\text{rand}}$ maintains overall scene fidelity, while $\mathcal{G}_{\text{uncertain}}$ provides targeted supervision to gradient-deficient areas.

We compute two rendered images: $I_{\text{full}}$ using the full set of Gaussians $\mathcal{G}$, and $I_{\text{sup}}$ using the uncertainty-intensified subset $\mathcal{G}_{\text{sup}}$. Each image is supervised independently against the ground-truth image $I_{\text{gt}}$ using $\ell_1$ and SSIM losses:

$$\mathcal{L} = \sum_{i \in \{\text{full}, \text{sup}\}} \lambda_i \left[ \mathcal{L}_{\text{rgb}}(I_i, I_{\text{gt}}) + \mathcal{L}_{\text{ssim}}(I_i, I_{\text{gt}}) \right], \tag{6}$$

where $\lambda_{\text{full}} + \lambda_{\text{sup}} = 1$, and we set both to $0.5$ in practice. We denote the losses for the full set render $I_{full}$ and the uncertainty-intensified subset render $I_{sup}$ as full loss ($\mathcal{L}_{full}$) and subset loss($\mathcal{L}_{sup}$), respectively. The uncertainty-intensified rasterization strategy is conceptually inspired by Dropout (Park et al., 2025; Srivastava et al., 2014) and Hard Negative Mining (Xuan et al., 2020; Jang et al., 2019). Randomly sampling $\mathcal{G}_{\text{rand}}$ provides stochastic diversity, allowing weakly contributing Gaussians to be supervised when dominant ones are excluded. Meanwhile, the deterministic inclusion of $\mathcal{G}_{\text{uncertain}}$ ensures consistent gradient flow to Gaussians that are persistently under-optimized. This dual mechanism reinforces learning in uncertain or ambiguous regions without modifying the rasterization process, and complements full-image supervision to maintain global photometric fidelity.

| Category | Methods | PSNR↑ | SSIM↑ | LPIPS↓ |
|---|---|---|---|---|
| Rule-based | Random | 19.969 | 0.584 | 0.456 |
| | ACP | 20.325 | 0.596 | 0.449 |
| 2D-based | MUSIQ | 19.850 | 0.575 | 0.466 |
| | CrossScore | 21.076 | 0.612 | **0.448** |
| 3D-based | FisherRF | 20.642 | 0.595 | 0.450 |
| | Ours | **21.410** | **0.613** | 0.451 |

(a) Mip-NeRF 360 dataset

| Category | Methods | PSNR↑ | SSIM↑ | LPIPS↓ |
|---|---|---|---|---|
| Rule-based | Random | 24.847 | 0.893 | 0.117 |
| | ACP | 22.718 | 0.855 | 0.138 |
| 2D-based | MUSIQ | 25.237 | 0.889 | 0.119 |
| | CrossScore | 23.746 | 0.868 | 0.130 |
| 3D-based | FisherRF | 25.190 | 0.892 | 0.116 |
| | Ours | **26.580** | **0.907** | **0.110** |

(b) NeRF Synthetic dataset

| Category | Methods | PSNR↑ | SSIM↑ | LPIPS↓ |
|---|---|---|---|---|
| Rule-based | Random | 18.918 | 0.694 | 0.390 |
| | ACP | 19.604 | 0.711 | 0.377 |
| 2D-based | MUSIQ | 18.541 | 0.686 | 0.403 |
| | CrossScore | 19.709 | **0.725** | **0.366** |
| 3D-based | FisherRF | 19.455 | 0.710 | 0.381 |
| | Ours | **20.060** | 0.722 | 0.377 |

(c) Deep Blending & Tank and Temples dataset

Table 1: **Quantitative results for the Active View Selection**. We compare our model with (1) Rule-based models (Random, ACP (Kopanas and Drettakis, 2023)), (2) 2D-based models (MUSIQ, CrossScore (Chen et al., 2024)), and 3D-based models (FisherRF (Wen et al., 2024)). Results are averaged over 9 scenes from the Mip-NeRF 360, 7 scenes from NeRF-synthetic dataset and 7 additional scenes from the Deep Blending and Tanks and Temples. We conduct four trials for each scene and report average scores. For statistics for all trial please refer to Appendix E

By supervising both the full and uncertainty-intensified images, we promote stronger gradient flow toward uncertain or low-opacity Gaussians without compromising photometric quality. This strategy mirrors the role of residual skip connections in ResNet (He et al., 2016) (Fig. 4(b)), supporting more stable convergence and mitigating overfitting in sparse or wide-baseline training settings. It is particularly effective in the early stages of next-best-view selection, where reconstruction is sensitive to both sparsely initialized regions and supervision bias caused by overfitting to limited views.

## 4 EXPERIMENTAL RESULTS

**Dataset.** We evaluate our approach on two benchmark dataset: NeRF-Synthetic Mildenhall et al. (2020), and Mip-NeRF 360 (Barron et al., 2022). While both datasets comprises from synthetic object-scale to real-world outdoor scenes with full 360-degree coverage, its uniform and curated camera trajectories provide limited challenge for active view selection, since even simple heuristics (e.g., furthest-distance selection) already perform reliably under balanced coverage (Xiao et al., 2024). To address this limitation, we carefully curate an extended benchmark dataset inlucindg seven diverse scenes from Deep Blending (Hedman et al., 2018) and Tanks and Temples (Knapitsch et al., 2017), which introduce unbalanced view distributions and varied scene scales that better reflect practical conditions. All experiments are conducted using images at their original resolutions, and further details on dataset curation are provided in Sec. A of the Appendix.

**Counterparts.** We compare our method quantitatively and qualitatively against several active 3DGS baselines which operate solely on RGB images: FisherRF (Wen et al., 2024), ACP (Kopanas and Drettakis, 2023), and random view selection. We also included 2D based view selection methods adopted from Active View Selector framework (Wang et al., 2025b). Following this framework, we incorporated two image quality assessment (IQA) models (MUSIQ and CrossScore) to evaluate perceptual quality. Both models were re-implemented according to the authors' official instructions and publicly available code.

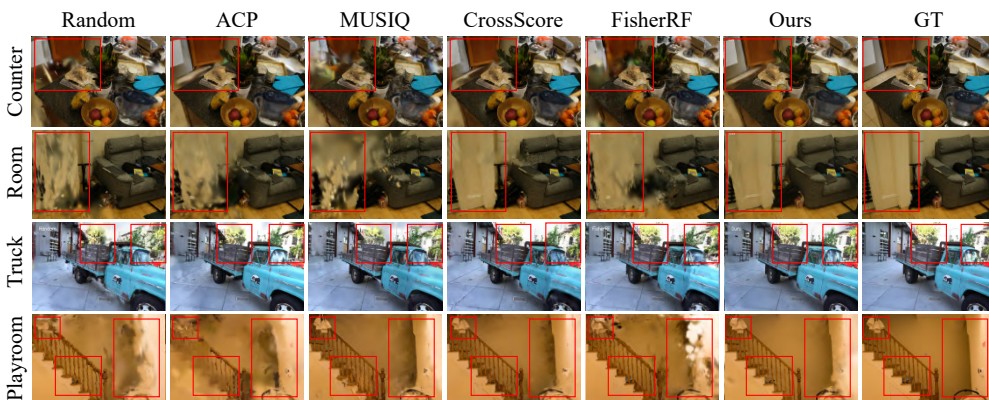

Figure 5: **Qualitative Comparison of Active View Selection.** Reconstruction from 20 selected views per scene. Our method shows improved completeness and fewer artifacts compared to baselines. For multi-view visualization, please refer the Appendix. Sec. C and supplementary video.

## 4.1 ACTIVE VIEW SELECTION

**Experimental settings.** Following the experimental protocols outlined by Wen et al. (2024), we adopt their prescribed initial view configurations and view selection schedules. Specifically, our experiments initiate with four uniformly distributed views, subsequently selecting an additional view every 100 epochs until reaching a total of 20 training views (For fewer training views, see Appendix Sec. C). We apply the same active selection strategy consistently across all datasets. For consistency, each model is initialized with the same random seed and trained for 20,000 iterations. All other settings remain unchanged across experiments, with exception of the view selection algorithms.

**Results.** Quantitative and qualitative results for the Mip-NeRF 360 dataset and additional scenes from the Deep Blending and Tank and Temples datasets are summarized in Table 1 and Fig. 5. The counterparts exhibit limited performance in 3D reconstruction, particularly in regions with sparse observations, due primarily to biased view selection and overfitting caused by vanishing gradients. This results in incomplete reconstructions, characterized by floating artifacts and missing geometry, *i.e.* holes and missing objects. In contrast, our method consistently delivers improved reconstruction quality, ranking first in PSNR and SSIM metrics and second in LPIPS for the Mip-NeRF 360 dataset. Our uncertainty-guided residual learning approach, based on dropout effects, produces smoother reconstructions in uncertain regions without compromising comparative performance. For camera view distribution, see Appendix Sec. B.

Additionally, experimental outcomes on the additional datasets validate the generalizability of our method. The Deep Blending dataset demonstrates our method's effectiveness in handling real-world-like diverse camera distributions. Likewise, the extensive outdoor scenes of the Tank and Temples dataset further illustrate our enhanced coverage, exemplified by the Truck scene in Figure 5.

## 4.2 COMPARISON ON UNCERTAINTY ESTIMATION

**Experimental settings.** We evaluate the effectiveness of our method in improving uncertainty estimation accuracy. Specifically, we examine whether incorporating residual loss ([‡]+ResGS) and self-augmented prefiltering ([†]+SA-ResGS) enhances the alignment between depth errors and predicted uncertainties under otherwise identical conditions. To measure this, we utilize the Area Under the Sparsification Error (AUSE) metric—a standard for evaluating uncertainty calibration adopted in (Wen et al., 2024; Goli et al., 2024; Shen et al., 2022)—where lower scores indicate better alignment between uncertainty and actual error. Thus, a lower AUSE score indicates better alignment between uncertainty predictions and actual errors, reflecting superior uncertainty calibration.

Following the approach used in CF-NeRF Shen et al. (2022), we employ depth maps from the NerfingMVS Wei et al. (2021) network, optimized using stereo depth from COLMAP at test time. Experiments are conducted on the all nine scenes from Mip-NeRF 360 under identical view selection and evaluated using all test views.

| Methods | Our proposed methods | | Metrics | | |
|---|---|---|---|---|---|
| | Sec. 4.2 | Sec. 4.4 | PSNR | SSIM | LPIPS |
| FisherRF[†] | - | - | 20.814 | 0.603 | 0.452 |
| [‡]+ResGS | - | ✓ | 21.022 | 0.596 | 0.459 |
| [†]+ResGS | - | ✓ | 20.740 | 0.610 | **0.442** |
| [†]+SA-HashGS | ✓ | - | 21.051 | 0.608 | 0.450 |
| [†]+SA-ResGS | ✓ | ✓ | **21.325** | **0.610** | 0.450 |

Table 2: **Ablation using Mip-NeRF 360 dataset.** [‡] denotes fixed-order view selection, replicating the original selection sequence from FisherRF[†], whereas [†] indicates dynamically updated view selection based on the model's progressive training status.

**Results.** We compare the average AUSE among 9 scenes in Mip-NeRF 360. By incorporating residual learning applied in ResGS, we observed a reduction in AUSE from 0.327 to 0.323, and subsequently to 0.297, when progressing from baseline (FisherRF[†]) to [‡]+ResGS (adding residual loss), and and [‡]+SA-ResGS (adding self-augmented prefiltering) respectively. These reductions demonstrate that both residual learning and self-augmented prefiltering enhance uncertainty calibration. The re-

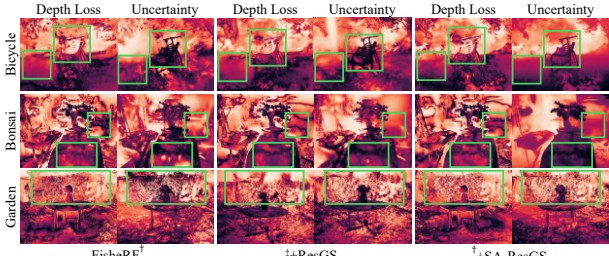

Figure 6: **Uncertainty Comparison.** We visualize depth loss with their corresponding uncertainty. Highlighted green boxes show that +SA-ResGS yields better alignment between large depth errors and high uncertainty

sults signify enhanced alignment between predicted uncertainty and actual depth errors (see Fig. 6). This improvement is attributed to the ResGS approach, which increases certainty in regions with low prediction error while enabling continued learning through skip connections in uncertain regions. Consequently, our model achieves both structurally and quantitatively superior uncertainty calibration, leading to improved accuracy in uncertainty-driven tasks such as active mapping.

## 4.3 ABLATION STUDIES

**Effect of individual components.** The ablation study in Table 2 highlights the contributions of each component in our proposed SA-ResGS framework. Incorporating residual learning (ResGS) alone improves reconstruction stability, particularly in ambiguous or sparse regions ([‡]+ResGS). However, it shows limitations when view selection remains uncontrolled ([†]+ResGS), indicating that training improvements alone are insufficient, especially under high computational uncertainty quantification errors. As shown in Sec. 4.2, our training module effectively aligns predicted uncertainty with actual losses, yet purely uncertainty-driven view selection strategies remain vulnerable to bias from internal learning dynamics.

In contrast, self-augmented prefiltering (SA-HashGS) independently improves geometric coverage through physically grounded view selection. Notably, combining SA-HashGS with ResGS yields substantial synergistic improvements, confirming the complementarity between robust view selection and residual supervision in optimizing both uncertainty quantification and reconstruction quality. For visual comparison please see Appendix Sec. D.

**Effect of the full loss.** We compare our method with a variant that updates Gaussians only through guided subset (Fig. 7). Without the full loss, the model often over-smooths ambiguous or low-confidence regions, causing high-frequency detail loss. This occurs because rule-based updates in 3DGS, such as pruning, remove Gaussians that lack sufficient gradients. Without the main reconstruction loss reinforcing these regions, the subset-only variant lacks a mechanism to preserve or re-activate Gaussians that require continued refinement. The full ResGS, by integrating residual and full losses, prevents under-updated Gaussians from collapsing and preserves both global structure and fine details. For ablation on uncertainty-guided sampling please refer Appendix D, Fig. S8.

Figure 7: **Comparison of novel-view renderings with and without the full loss in ResGS.** The w/o full loss ($\mathcal{L}_{sup}$) variant exhibits smoothing artifacts, whereas incorporating the full loss ($\mathcal{L}_{full} + \mathcal{L}_{sup}$) preserves scene structure and high-frequency details. Note that both employ fixed-order view selection, using pre-selected view-order from FisherRF[†] to ignore view selection effect.

**Robustness to correspondence noise.** SA-Points are obtained through triangulation of dense correspondences, and their accuracy can influence reconstruction and view selection. To examine robustness, we conducted experiments with synthetic correspondence noise (0.0–10.0 pixels). The system remains stable up to moderate noise levels ($\leq$ 5.0 pixels) when combined with a 1-pixel reprojection filter, as shown in Table. 3. Larger perturbations cause noticeable degradation, indicating that the system tolerates moderate correspondence errors without significant performance drop.

| Noise | 0.0 | 0.5 | 1.0 | 5.0 | 10.0 |
|-------|-----|-----|-----|-----|------|
| PSNR  | 24.441 | 24.199 | 24.117 | 24.311 | 23.121 |

Table 3: **Robustness to synthetic correspondence noise.** PSNR remains stable under moderate perturbations ($\leq$ 5 pixels), showing SA-Points are resilient to realistic levels of noise.

## 4.4 Computation Efficiency Analysis

A key challenge in active view selection is computational cost, as FisherRF computes per-Gaussian Fisher information via backpropagation across all candidate views, creating bottlenecks in large-scale datasets. To evaluate our prefiltering strategy, we conducted a runtime analysis on the Bonsai scene using a mid-range GPU (38 TFLOPS fp32), summarized in Table 4. SA-ResGS replaces exhaustive Fisher evaluation with a four-step process: dense correspondence prediction (MASt3R), triangulation for SA-Points, voxel-based prefiltering, and Fisher computation on only 20% of views. Despite these extra steps, view selection is 55% faster (28.0s $\rightarrow$ 12.5s), with only a modest increase in per-iteration cost (0.005s $\rightarrow$ 0.027s). Unlike FisherRF, whose cost grows with candidate views and becomes impractical for large datasets such as ScanNet (about 1600 images), SA-ResGS maintains fixed training cost once the schedule is set. GPU memory usage rises slightly but remains within standard limits, making our approach both scalable and practical. We further report the end-to-end active-reconstruction runtime under same setting. In five repeated run, SA-ResGS reduced runtime by 55%. This aligns with Table 4: the reduced view-selection cost outweighs the extra operations.

| Process | MASt3R | Triangulation | prefilter | Fisher | Total | Raster. | GPU | End-to-end |
|---------|--------|---------------|-----------|--------|-------|---------|-----|------------|
| FisherRF | — | — | — | 28.00 s | 28.00 s | 0.005 s/iter | ~8K | 32 m 59 s |
| Ours | 0.40s | 1.49 s | 5.00 s | 5.60 s | 12.50 s | 0.027 s/iter | ~10.5K | 19 m 45 s |

Table 4: **Runtime comparison of FisherRF and SA-ResGS on the Bonsai scene.** Breakdown of view selection and training costs. SA-ResGS introduces additional prefiltering steps but reduces total runtime by 55%, with only modest increases in per-iteration cost and GPU memory usage.

## 5 Conclusion

This paper presents SA-ResGS, a Self-Augmented Residual 3D Gaussian Splatting framework for stabilize uncertainty quantification and enhance uncertainty-aware supervision in next-best-view (NBV) selection for active scene reconstruction. We introduce Self-Augmented Points, triangulated from a training view and a rasterized extrapolated view. These points enable physically grounded view selection and help mitigate erroneous uncertainty bias, while supporting targeted 3D supervision in high-uncertainty regions. Furthermore, we propose the first residual learning strategy tailored to 3D Gaussian Splatting, enabling effective supervision for both uncertain image regions and weakly contributing Gaussian splats. This leads to improved photometric reconstruction in novel view synthesis. Experimental results demonstrate the effectiveness of SA-ResGS across a range of realistic scenes, encompassing both indoor and outdoor environments and varying scene scales.

## LIMITATIONS AND FUTURE WORK

SA-ResGS has several limitations. The method also relies on hyperparameter choices, potentially affecting performance to specific scenarios. Method to probabilistically choosing parameter would have potential to handle such limitations. We believe that applying residual supervision intermittently or adopting a tighter prefiltering ratio—without compromising reconstruction quality can further reduce computational cost. Current method relies on 2D feedforward matching models, which show robust correspondence matching in most cases, application with more stronger model would further enhance the performance of the proposed methods. Extending SA-ResGS to dynamic environments involving moving objects or illumination changes remains an open challenge.

## ETHIC STATEMENT

Our work focuses on advancing active 3D scene reconstruction through residual supervision and physically guided view selection in 3D Gaussian Splatting. The proposed method is entirely built upon publicly available datasets such as MipNeRF360 (Barron et al., 2022), Deep Blending (Hedman et al., 2018), and Tank and Temples dataset (Knapitsch et al., 2017), which are widely used in the vision community. No personally identifiable or sensitive information is included in our experiments. However, as with all vision datasets, there may exist inherent biases in scene distributions (e.g., indoor environments being more common than outdoor scenes). These biases could affect generalization in certain deployment scenarios. We encourage users to be aware of such limitations when applying our method to broader real-world applications.

## RREPRODUCIBILITY STATEMENT

We provide detailed descriptions of our algorithmic components in Sec. 3, including uncertainty-guided supervision, view selection strategies, and residual learning for 3DGS. The datasets used (MipNeRF360 (Barron et al., 2022), DeepBlending (Hedman et al., 2018), Tanks and Temples datasets (Knapitsch et al., 2017)) and the evaluation protocols are specified in Sec. 4. To facilitate reproducibility, the hyperparameters, training recipes, and implementation details are documented in Appendix B. All experiments were conducted with standard PyTorch libraries and commonly available GPUs, ensuring that the results can be reproduced with reasonable computational resources.

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

## APPENDIX OVERVIEW

This supplementary document provides additional implementation details and results that support and extend the main paper. It is organized as follows:

## A    IMPLEMENTATION DETAILS

### A.1    COVERAGE ESTIMATION

We follow the baseline setup of FisherRF (Wen et al., 2024), initializing our experiments with a sparse point cloud reconstructed via COLMAP (Schonberger and Frahm, 2016; Schonberger et al., 2016). This SfM output defines the axis-aligned bounding volume for discretizing the scene into voxel grids. To enable physically grounded view selection, we construct Self-Augmented Points (SA-Points) via triangulation between a training view and its extrapolated view. SA-Points are used to estimate surface occupancy and to encode observed geometry into binary voxel features. Candidate views are scored based on their voxel-level dissimilarity to the encoded training views, supporting robust coverage estimation without relying on early uncertainty signals.

In the following, we detail (1) Self-Augmented Points (SA-Points) Generation, and (2) Observed Surface Coverage Estimation and View Frustum Construction.

#### A.1.1    SELF-AUGMENTED POINTS (SA-POINTS) GENERATION

**Extrapolated View Generation.** To synthesize a novel rasterized view while maintaining sufficient scene overlap, we perturb the original camera center by $\pm 0.25$ units along the x and y axes and translate it backward by 0.5 units along the z-axis. This backward-only perturbation ensures that the extrapolated view retains a high degree of visibility overlap with the original training view, keeping most scene content within the shared frustums. As a result, the extrapolated view covers a large portion of the original training image while still providing a novel perspective of the same surfaces, enabling reliable correspondence estimation for SA-Point generation.

**Dense Correspondence Matching.** We compute dense correspondences between the ground truth image of original training view and the rendering from extrapolated view using the pretrained MASt3R model (Leroy et al., 2024) (MASt3R_ViTLarge_BaseDecoder_512_catmlpdpt_metric), which is robust to moderate viewpoint perturbations. As discussed in the main paper, this 3D vision foundation model produces context-aware dense correspondences by capturing structured scene semantics over 16×16 local patches, even when the extrapolated view includes geometric artifacts. This robustness allows us to extract reliable matches despite distortions in the rasterized extrapolated images, enabling consistent 3D reconstruction from single-view observations. Consequently, we can leverage extrapolated viewpoints to augment sparse training views without requiring multiview supervision.

**Triangulation and Reprojection Filtering.** To ensure geometric consistency, we triangulate 3D points from matched correspondences and filter them based on reprojection error, discarding points with a bidirectional reprojection error exceeding 0.5 pixels. For computational efficiency, we parallelize the triangulation process across multiple CPU threads and apply a spatial stride of 5 pixels in both x and y directions to subsample correspondence pairs. These filtering strategies significantly accelerate processing while preserving high-fidelity geometric structure.

### A.1.2 Observed Surface Coverage Estimation and View Frustum Construction

**Voxel Grid Construction.** Given the sparse SfM points, we define an axis-aligned bounding box (AABB) that encapsulates the entire 3D point cloud. The AABB is computed from the minimum and maximum bounds of the reconstructed points.

**Initial Occupancy Estimation.** We discretize the scene into a voxel grid and mark a voxel as occupied if it contains a minimum number of SfM points. To better represent scene geometry, we apply $N$-fold upsampling to the occupied voxels. For the minimum number for determining occupancy, we use 2 for outdoor scenes, and 5 for indoor cases.

**Observed Region Calculation.** SA-Points are mapped to their nearest voxels to define the observed surface region. To account for possible triangulation errors and improve spatial robustness, we apply a 3D dilation operation to the occupied voxels. For all cases, a dilation radius of 2 is applied.

**View Frustum Determination.** To evaluate candidate views, we define view frustums using camera intrinsics and the global maximum bounds computed from the SfM point cloud. These frustums ignore visibility constraints but serve as a conservative estimate of potential scene coverage. Coverage scores for physically grounded view selection are then computed by measuring voxel-level intersections between the candidate frustums and the observed surface, represented via hash-encoded voxel occupancy representation.

## A.2 Experimental setup

All models presented in the main manuscript, including the ablation variants of our proposed method, were trained and evaluated on a single NVIDIA V100 GPU with 32 GB of memory. CPU-based components—such as SA-Points generation, voxel grid processing, and triangulation—were parallelized across 8 threads for efficiency. Each experiment was run once with a fixed random seed of 0 to ensure reproducibility.

### A.2.1 Dataset Detail

We evaluate our method and baselines on three types of datasets: (1) Mip-NeRF 360, (2) NeRF-Synthetic and (3) Extended NBV benchmark datasets. Mip-NeRF 360 consists of nine real-world scenes with dense 360-degree camera coverage, captured across both indoor and outdoor scenarios. NeRF-Synthetic dataset consists of 8 object-centric, multi-view images with dense 360-degree camera view distribution, rendered via Blender. To ensure consistency across datasets, we generated sparse point clouds for all NeRF-Synthetic scenes using COLMAP. We note that Ficus scene fails to reconstruct reliably under COLMAP, therefore we excluded it and evaluated on the remaining scenes.

While this datasets provides a controlled and well-curated benchmark, its uniform view distribution limits the difficulty of next-best-view (NBV) evaluation. Such settings often make next-best-view (NBV) strategies appear less critical, since even simple heuristics (e.g., furthest-distance selection) already perform reliably under balanced coverage (Xiao et al., 2024)

To address this, we additionally construct an Extended NBV benchmark by selecting seven challenging scenes from Deep Blending and Tanks and Temples, characterized by irregular camera trajectories and diverse scene scales. This curated set introduces more realistic and unbalanced conditions, offering a complementary testbed for assessing robustness in active view selection.

These datasets contain large-scale and geometrically complex scenes with irregular camera distributions, but their difficulty also means that many methods fail outright. To filter such degenerate cases, we trained the FisherRF baseline under the standard scheme and retained only scenes where it achieved at least 17 dB PSNR, ensuring that the comparisons remained fair and informative.

This procedure produced seven representative scenes: Horse, Truck, Francis, Ballroom, Barn, Ponche, and Playroom. The selected set spans both indoor and outdoor environments, and includes highly complex camera distributions (e.g., Ballroom, Ponche, and Playroom) that deviate substantially from the curated coverage of Mip-NeRF 360. These characteristics create more realistic stress tests for NBV strategies by introducing occlusions, scale variations, and unbalanced observations. A brief visualization of scene configurations and camera trajectories is provided in Sec.2 of this supplementary material.

## B    CAMERA DISTRIBUTION ANALYSIS

To evaluate the effectiveness of our view selection strategy, we compare the camera distributions produced by different methods. Fig. S1, S3, and S1 visualize these distributions from both bird's-eye and side perspectives. As discussed in the main paper, FisherRF tends to produce clustered view selections due to its tight coupling with the internal 3DGS learning dynamics—visibly highlighted in the semi-transparent yellow regions. In contrast, our method yields a more spatially uniform and well-dispersed distribution of viewpoints.

This distinction is further illustrated in Fig. S4, where in the Room scene from the Mip-NeRF 360 dataset, FisherRF often selects redundant or near-parallel views. Our method instead promotes angular diversity, leading to broader scene exploration. A similar trend is observed in the Deep Blending dataset, where our approach selects viewpoints across a wider vertical range. These comparisons reinforce our claim that SA-ResGS facilitates physically grounded and geometrically diverse view selection, contributing to improved scene coverage.

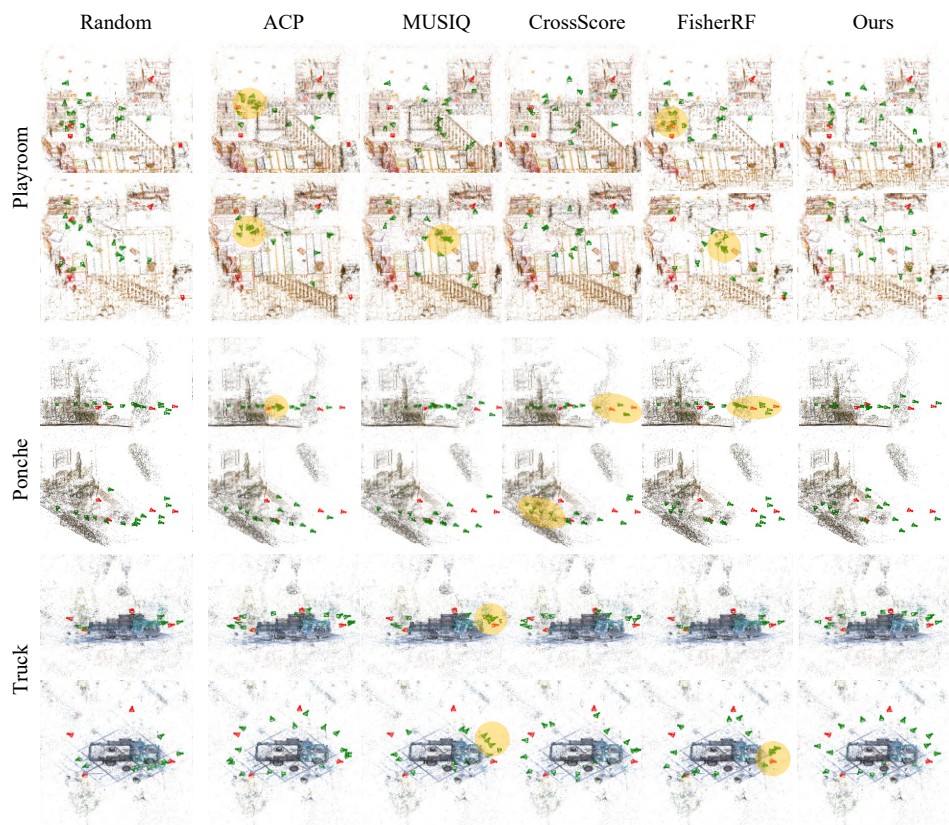

Figure S1: **Camera View Distribution on Deep Blending and Tanks & Temples** View distributions on additional datasets, following the same color and annotation scheme as Fig. S2. Our method maintains broader spatial coverage, while baseline methods often exhibit clustering (yellow circles), consistent with the biases observed in Mip-NeRF 360.

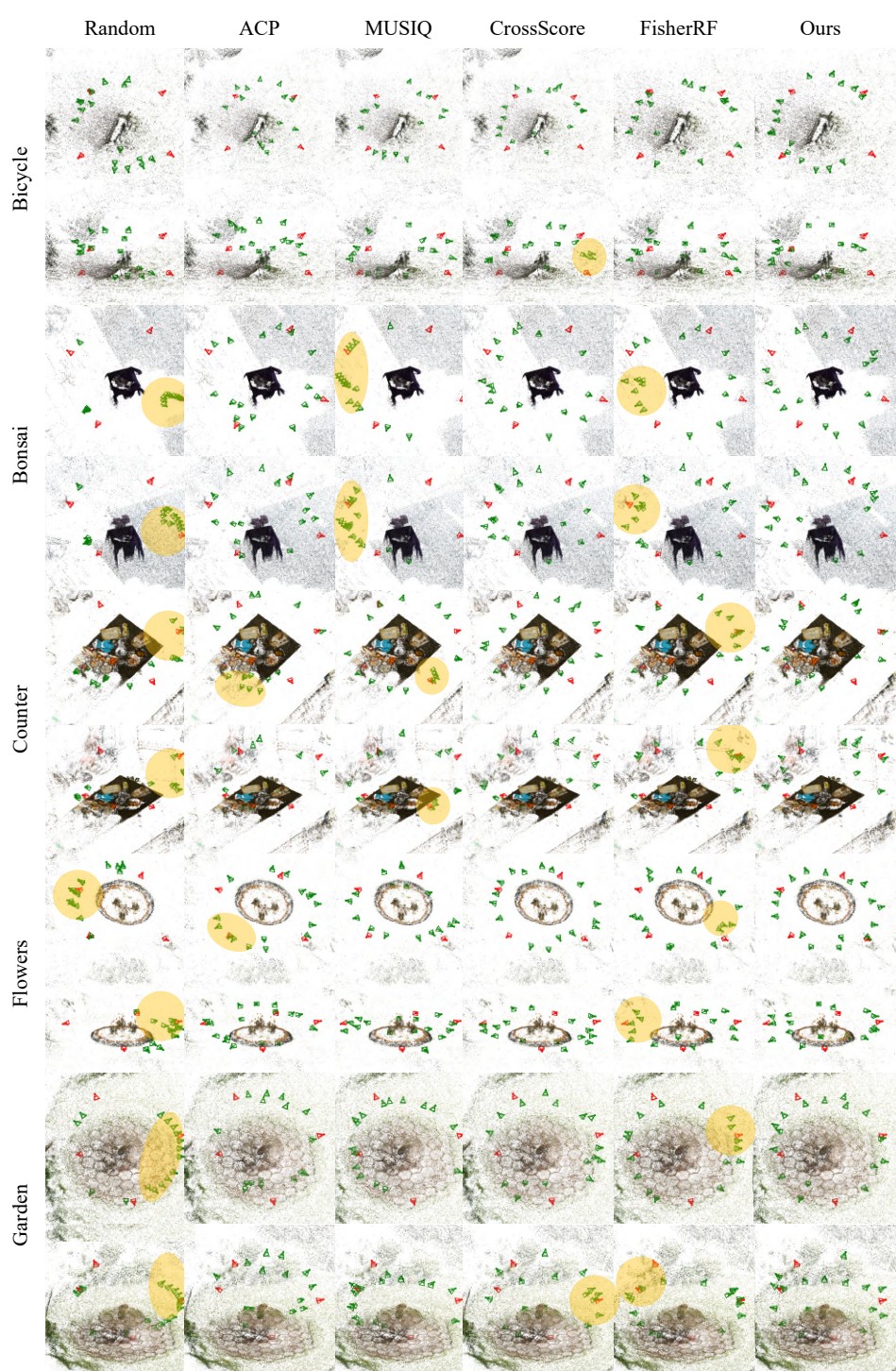

Figure S2: **Camera View Distribution on Mip-NeRF 360.** Visualization of camera poses selected by each method on the Mip-NeRF 360 dataset. Red frustums indicate the initial views, while green frustums denote views added during active selection. Our method produces a more uniformly distributed set of viewpoints, while both baselines exhibit clustered view selections (highlighted in yellow circle), particularly in FisherRF due to its reliance on uncertainty signals entangled with 3DGS training dynamics.

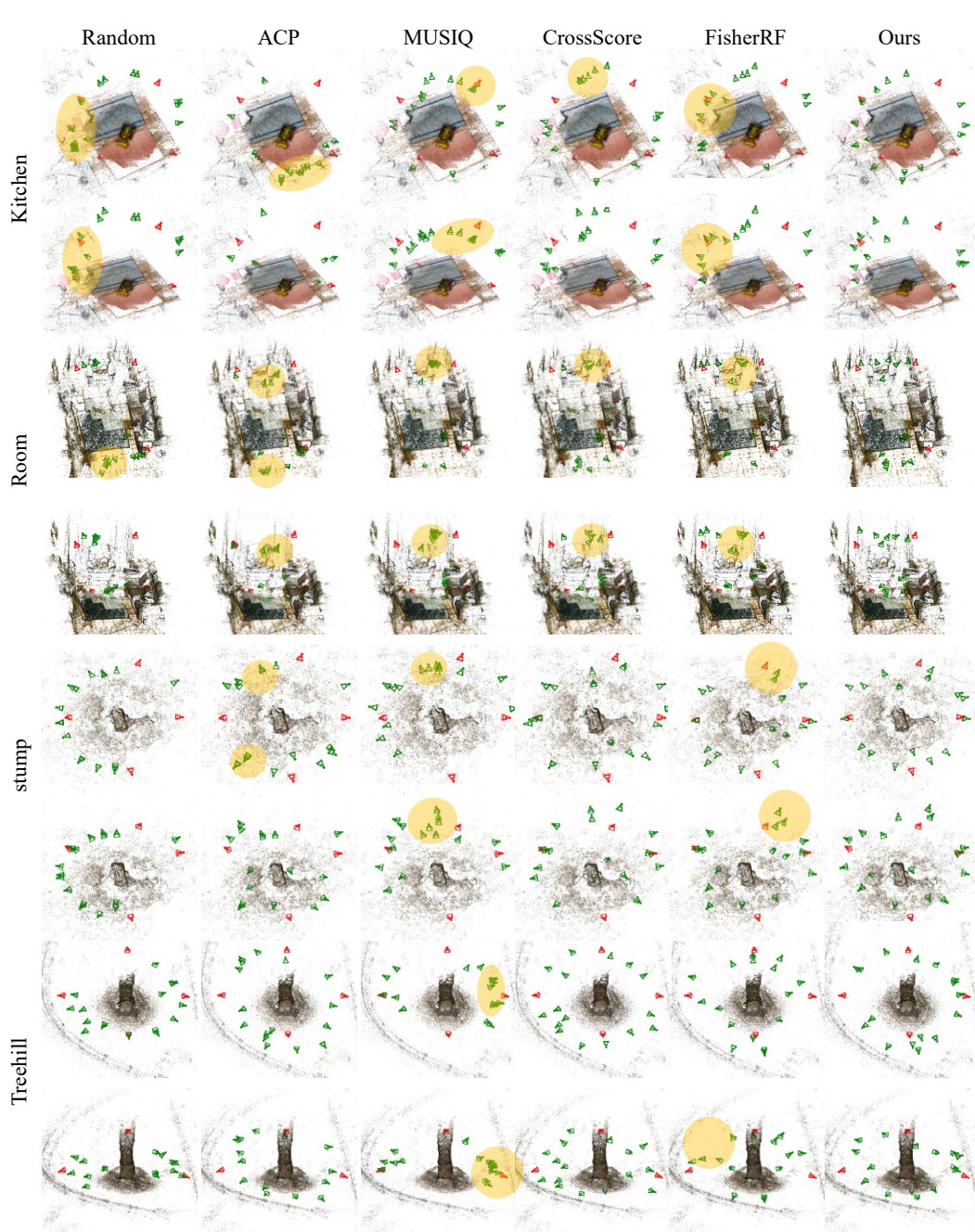

Figure S3: **Camera View Distribution on Mip-NeRF 360.** Visualization of camera poses selected by each method on the Mip-NeRF 360 dataset. Red frustums indicate the initial views, while green frustums denote views added during active selection. Our method produces a more uniformly distributed set of viewpoints, while both baselines exhibit clustered view selections (highlighted in yellow circle), particularly in FisherRF due to its reliance on uncertainty signals entangled with 3DGS training dynamics.

## C    EXTENDED QUALITATIVE COMPARISONS

The qualitative results presented in the main paper are limited by space constraints, which may obscure the full advantages of our method. To address this, we provide extended visualizations from

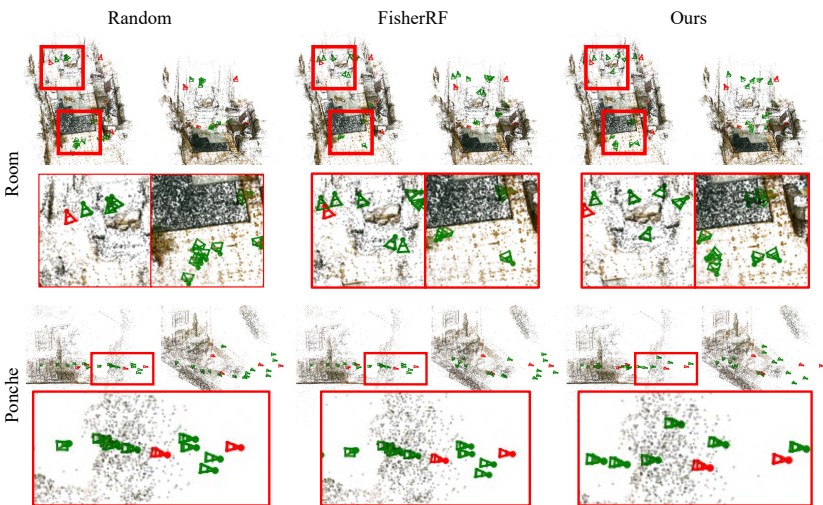

Figure S4: **Diversity of Selected Views (Zoomed-in Analysis)** We highlight the spatial diversity of selected camera poses by presenting zoomed-in regions corresponding to the boxed areas above. Compared to the baselines, our method selects views from a broader range of angles (Room, top) and elevations (Ponche, bottom), resulting in more comprehensive scene coverage.

multiple test viewpoints. Fig. S5a to S7 display results from five scenes across the Mip-NeRF 360, Deep Blending, and Tanks and Temples datasets, with six to eight novel test views per scene. Our method consistently achieves broader and more complete scene coverage compared to the FisherRF.

As discussed in the main paper, our residual supervision strategy further improves geometric consistency and reconstruction robustness, particularly in sparse or limited-view scenarios. This is especially beneficial under the standard protocol of active or next-best-view selection, where training begins with a small number of views (e.g., 4) and progressively adds new views (typically one at a time). The synergy between physically grounded view selection and residual learning enables high-fidelity reconstruction even from limited initial observations. As mentioned in Sec. F, we also include a supplementary video with 360-degree novel view renderings. We encourage reviewers to view this video to better appreciate the improvements in coverage and structural accuracy provided by our proposed SA-ResGS.

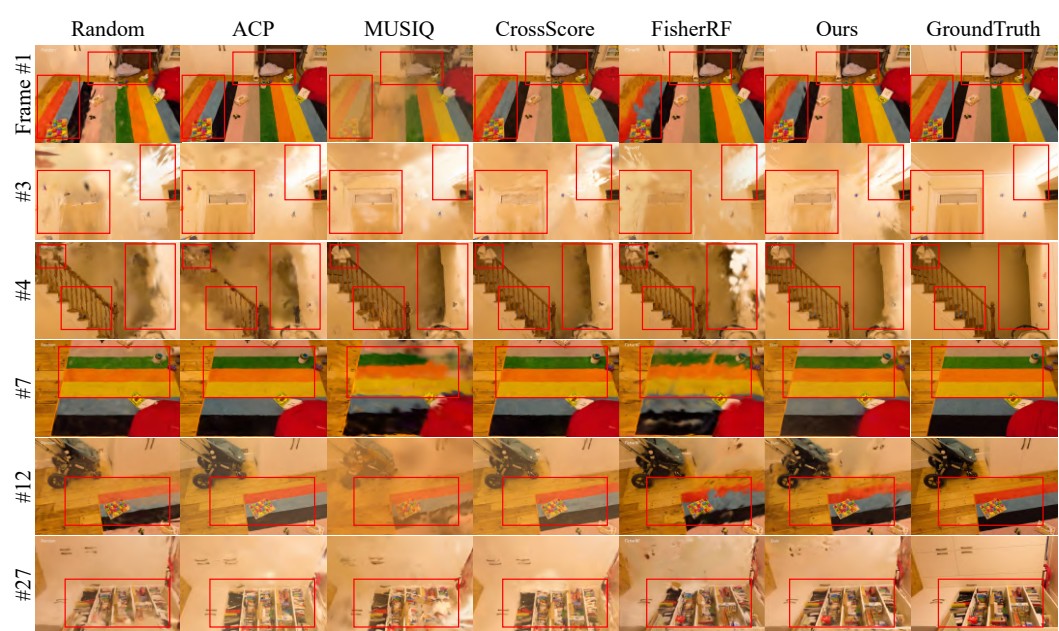

(a) **Playroom scene. Six test-time novel views.**

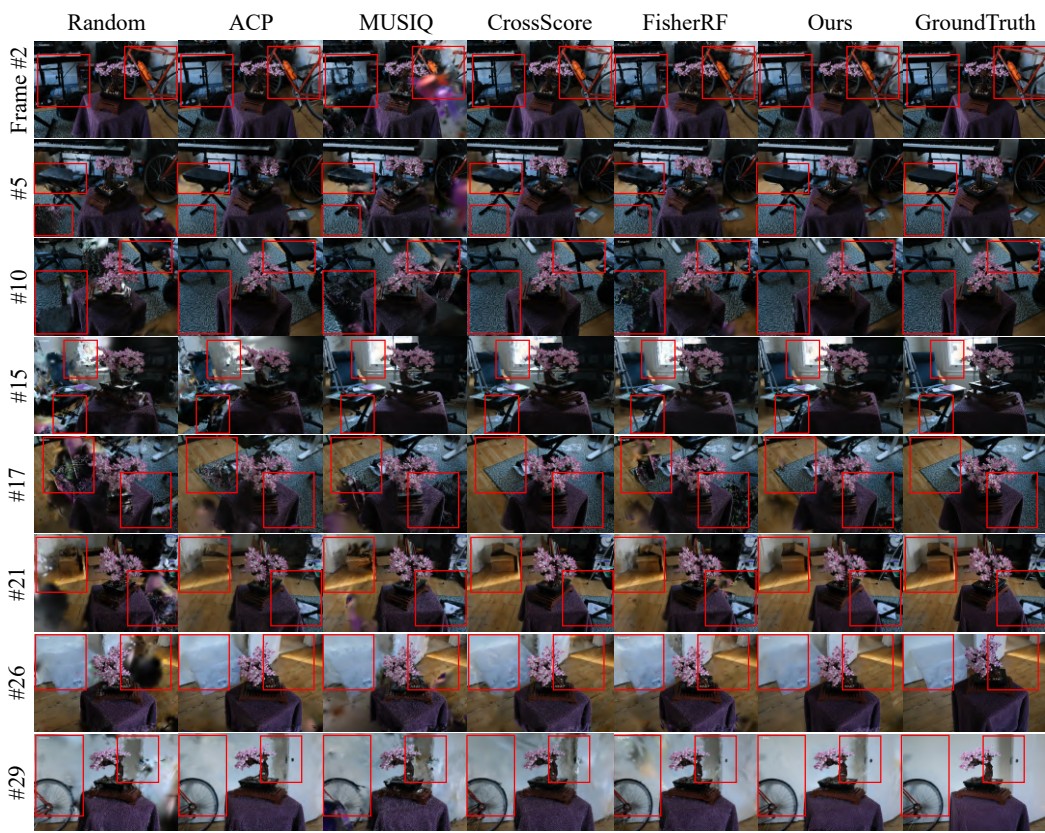

(b) **Bonsai scene. Eight test-time novel views.**

Figure S5: **Qualitative comparison across multiple test-time novel views.** Across all scenes, our method produces more complete and consistent reconstructions, particularly in **occluded or sparsely observed regions** (red boxes). In contrast, baseline methods (Random, FisherRF) often exhibit **missing geometry, blurring, or structural artifacts** due to biased or clustered view selection.

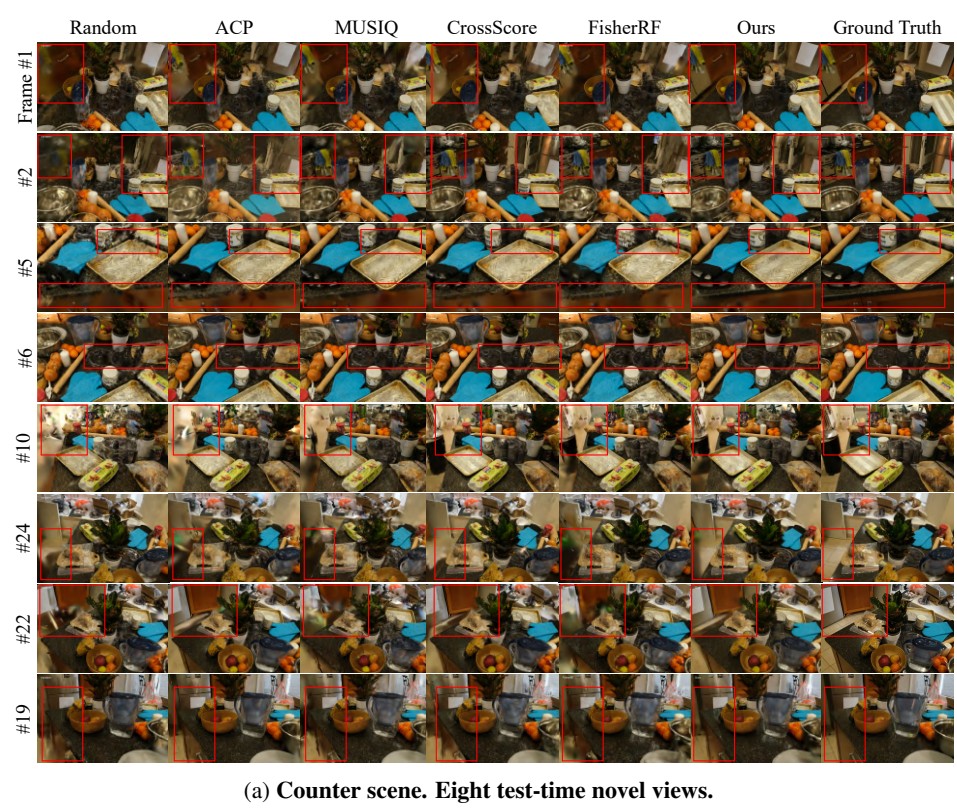

(a) **Counter scene. Eight test-time novel views.**

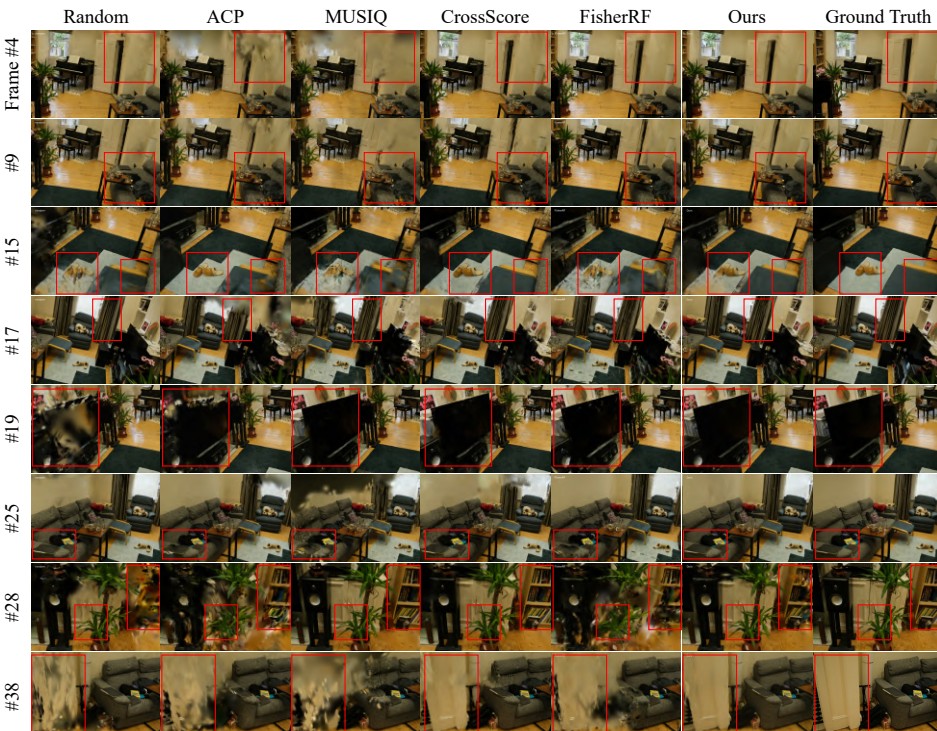

(b) **Room scene. Eight test-time novel views.**

Figure S6: **Qualitative comparison across multiple test-time novel views.** Across all scenes, our method produces more complete and consistent reconstructions, particularly in occluded or sparsely observed regions (red boxes). In contrast, baseline methods (Random, FisherRF) often exhibit missing geometry, blurring, or structural artifacts due to biased or clustered view selection.

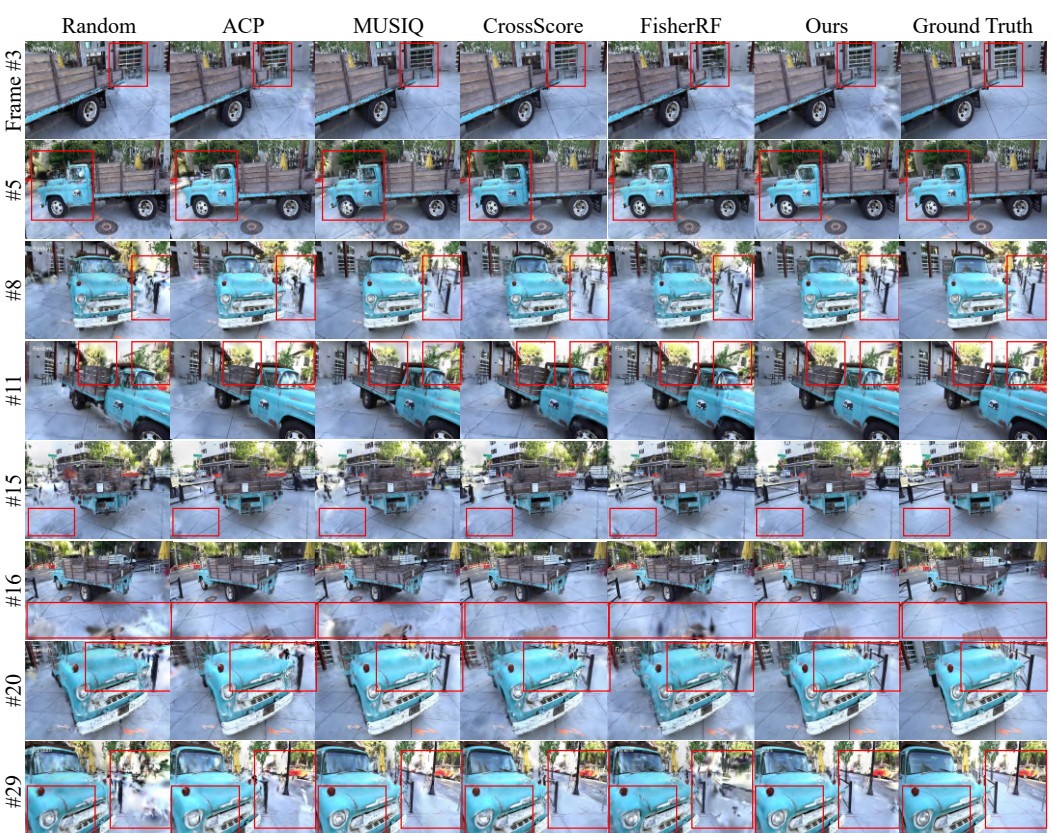

Figure S7: **Qualitative Comparison Across Multiple Test-Time Views (Truck Scene).** Across all scenes, our method produces more complete and consistent reconstructions, particularly in occluded or sparsely observed regions (red boxes). In contrast, baseline methods (Random, FisherRF) often exhibit missing geometry, blurring, or structural artifacts due to biased or clustered view selection.

# D    ADDITIONAL RESULTS FOR ABLATION STUDIES

**Qualitative Results for Ablation studies.** To supplement the quantitative results presented in the ablation study, we provide extended qualitative comparisons in Fig. S9, evaluating different model variants on the **Room** and **Counter** scenes. The variants include the baseline FisherRF, residual supervision with fixed-order view selection ($^\ddagger$+ResGS), our physically grounded view selection method (SA-HashGS), and the full SA-ResGS model combining both components. In the **Room** scene, the baseline FisherRF exhibits missing geometry and artifacts near occluded regions and object boundaries (orange boxes). SA-HashGS mitigates these issues by selecting geometrically diverse viewpoints, leading to improved surface coverage. Residual supervision ($^\ddagger$+ResGS) further refines local details—even in regions already observed by the baseline—demonstrating enhanced reconstruction without additional view coverage, as evidenced by improved toy geometry and floor textures (red boxes). The full SA-ResGS model combines both benefits, producing reconstructions that closely align with ground-truth in terms of both structural completeness and fine detail.

In the **Counter** scene, residual supervision ($^\ddagger$+ResGS) reduces floating artifacts and improves under-optimized regions caused by occlusion, such as the shadow-like area on the left side of the countertop (first row). SA-HashGS enhances global coverage, alleviating severe blurring and recovering geometry in previously unobserved areas (red boxes). The full SA-ResGS model integrates both benefits, yielding sharper and more complete reconstructions by jointly leveraging physically grounded view selection and residual learning. These results validate the complementary roles of physically grounded view selection and uncertainty-guided residual learning. Their integration in SA-ResGS consistently improves reconstruction fidelity under sparse-view settings.

Additional 360-degree renderings for these ablation results are included in the supplementary video and are recommended for further comparison.

**Ablation study on Residual Learning.** To further analyze the effect of the residual loss in ResGS, we conduct an expanded ablation study in Sec 4.3, as presented in Fig. S8. We compare three configurations: (a) w/ full loss, and w/ subset loss only (b) w/ full Loss, and w/ subset loss using random sampling only, and (c) w/ full Loss, and w/ subset loss using both uncertainty-guided sampling and random sampling For fair comparison, the sampling ratio in (b) and (c) is matched by increasing the amount of random sampling in (b). To separate the effect of view sampling, we employ fixed-order view selection, replicating the original selection sequence from FisherRF[†], to isolate and examine the effect of residual loss.

Consistent with the observations in the main paper, the absence of the full loss (a) results in pronounced over-smoothing, particularly in ambiguous or low-confidence regions. Moreover, when the supervised loss relies solely on random sampling (b), structural degradation becomes evident in areas that frequently undergo occlusions—such as the staircase behind the door or the window-frame region. This occurs because random sampling alone cannot reliably propagate gradients to Gaussians that require sustained refinement.

In contrast, the proposed full configuration (c), which incorporates both uncertainty-guided and random sampling, consistently preserves fine details and maintains structural coherence. By enforcing continuous gradient flow to under-updated or geometrically unstable Gaussians—particularly those that are bulky, floating, or insufficiently activated—our approach effectively corrects misaligned geometry and enhances robustness in challenging regions. These results highlight the advantage of our residual-learning-based update mechanism, which yields qualitatively more faithful and geometrically consistent renderings across diverse novel viewpoints.

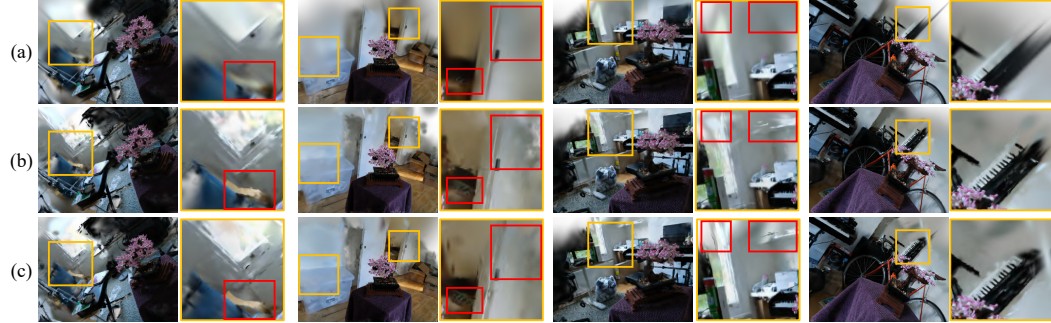

Figure S8: **Qualitative comparison across multiple test-time novel views.** (a) w/ subset loss only, (b) w/ full loss and subset loss using random sampling only, and (c) w/ full loss and subset loss using both uncertainty-guided and random sampling. For a fair comparison between (b) and (c), the overall sampling ratio is matched by increasing the random sampling rate in (b). Highlighted regions (red boxes) denote areas where differences between the variants become especially pronounced.

**Ablation study on extended datasets.** We additionally evaluated all ablation configurations on the Deep Blending and Tanks & Temples datasets to verify their generality, as shown in Table S1. Across scenes, ResGS consistently improves reconstruction under an identical view sequence, whereas combining ResGS with dynamic selection remains sensitive to early uncertainty noise. SA-HashGS stabilizes early decisions via surface-aware filtering, and the full configuration (SA-HashGS + ResGS) achieves the strongest and most stable improvements across datasets. Note that, the Barn scene was excluded due to repeated convergence failures across some ablation methods.

## E  ADDITIONAL ABLATION ANALYSIS

**Robustness to Correspondence Noise.** SA-Points are obtained through triangulation of dense correspondences, so their accuracy can influence reconstruction and view selection. We adopt

| Methods | Our proposed methods | | Metrics | | |
|---|---|---|---|---|---|
| | Sec. 4.2 | Sec. 4.4 | PSNR | SSIM | LPIPS |
| FisherRF[†] | - | - | 19.293 | 0.715 | 0.378 |
| [‡]+ResGS | - | ✓ | 19.462 | 0.719 | 0.381 |
| [†]+ResGS | - | ✓ | 19.887 | 0.730 | 0.374 |
| [†]+SA-HashGS | ✓ | - | 19.890 | 0.730 | 0.364 |
| [†]+SA-ResGS | ✓ | ✓ | 20.305 | 0.740 | 0.362 |

Table S1: **Ablation using Deep Blending & Tank and Temples datsaet.** [‡] denotes fixed-order view selection, replicating the original selection sequence from FisherRF[†], whereas [†] indicates dynamically updated view selection based on the model's progressive training status.

| Reprojection error (pixels) | PSNR | SSIM | LPIPS | Coverage |
|---|---|---|---|---|
| 0.5 | 22.634 | 0.817 | 0.332 | 57.74% |
| 1.0 | 24.441 | 0.838 | 0.317 | 94.11% |
| 2.0 | 24.244 | 0.834 | 0.318 | 94.89% |

Table S2: **Effect of reprojection filtering thresholds.** A 1-pixel threshold offers the best trade-off between accuracy and coverage, confirming that accurate filtering is essential for stable SA-Points.

reprojection error to filter out inconsistent points, to examine robustness, we conducted experiments with different reprojection thresholds.

Varying the reprojection error threshold further confirmed this robustness. As shown in Table. S2, very strict filtering (0.5 pixels) reduced coverage, while very loose filtering (2.0 pixels) introduced slight inaccuracies. A threshold of 1.0 pixel provided the best trade-off, ensuring sufficient coverage and reliable geometry.

These results demonstrate that SA-Points are resilient to moderate correspondence errors. Our combination of reprojection filtering and voxel dilation effectively balances coverage and accuracy, enabling stable and reliable NBV selection in practice.

**Effects of number of selected views.** The number of available input views is a critical factor in reconstruction quality, particularly under sparse-view settings. To analyze this effect, we conducted an ablation on the Bonsai scene, starting from 4 fixed initial views and incrementally adding views using our selection strategy. For clarity, we uniformly subsampled training views and report PSNR across different totals.

Results in Table S3 that under sparse conditions (7–10 views), our method reaches higher PSNR with fewer inputs compared to FisherRF, demonstrating the benefit of physically grounded filtering in stabilizing early-stage. As more views are added, the initial advantage narrows but new gains appear from 13 views onward, where residual learning further refines Gaussian parameters and improves geometry.

Overall, the method excels in low-view regimes while continuing to scale effectively with more observations, validating its robustness across varying view counts.

| Selected Views | 4 | 7 | 10 | 13 | 16 | 19 |
|---|---|---|---|---|---|---|
| random | 15.929 | 16.485 | 17.573 | 17.694 | 18.233 | 18.400 |
| FisherRF | 15.985 | 17.269 | 18.773 | 20.188 | 21.722 | 22.650 |
| Ours | 15.970 | 17.606 | 18.895 | 20.159 | 23.229 | 24.064 |

Table S3: **Performance scaling with the number of selected views.** Our method yields higher PSNR in sparse-view regimes (7–10 views) and continues to improve as more views are added, highlighting both early stability and long-term scalability.

**Ablation on Hash encoding size.** To assess the robustness of the hash-encoded voxel grid used in the coverage prefilter, we conducted an ablation study by varying the hash-table size from $2^{11} - 2^{19}$, as

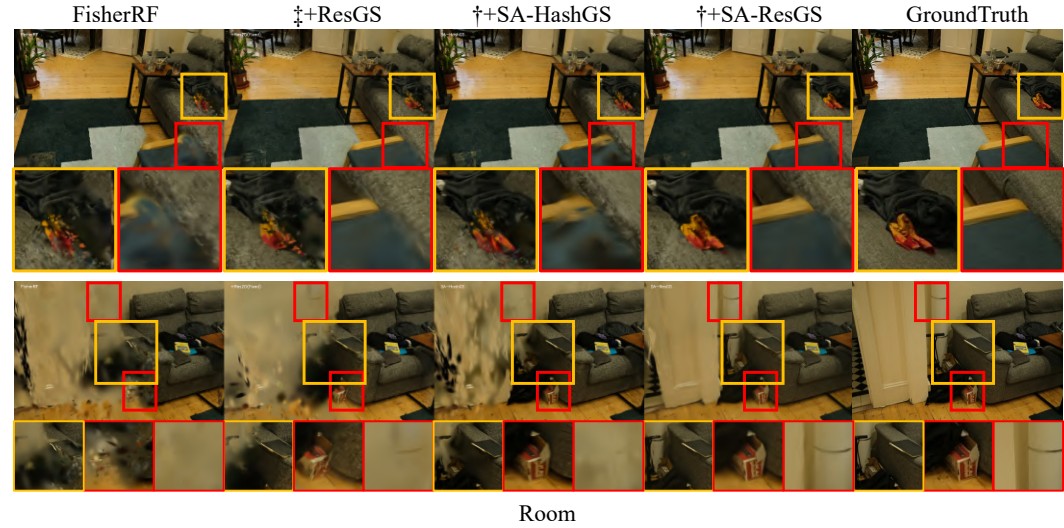

FisherRF ‡+ResGS †+SA-HashGS †+SA-ResGS GroundTruth

Room

FisherRF ‡+ResGS †+SA-HashGS †+SA-ResGS GroundTruth

Counter

Figure S9: **Qualitative Self-Comparison for Ablation Study.** Results on the **Room** and **Counter** scenes comparing FisherRF, ‡+ResGS, SA-HashGS, and SA-ResGS. Orange boxes highlight improvements from coverage-guided view selection, while red boxes emphasize the effects of residual supervision. Residual learning enhances geometric stability and reduces artifacts (e.g., jittering surfaces), whereas SA-HashGS recovers unobserved (e.g., occluded or shadowed areas). The full SA-ResGS combines both benefits, yielding the most complete and and structurally faithful reconstructions.

well as a no-collision variant implemented with direct indexing. Despite more than a 200× difference in hash capacity, the reconstruction accuracy and coverage estimation remained remarkably stable across all scenes, exhibiting no consistent trend of degradation, as shown in Table S4.

This insensitivity arises from two structural properties of our pipeline. (1) Only occupied voxels are hashed, while empty regions are skipped entirely; because occupancy varies widely across scenes, the effective load factor of the hash table remains low even for relatively small hash sizes. (2) Residual inconsistencies introduced by collisions are further mitigated by the subsequent Fisher-based fine selection stage, which provides an additional layer of error correction.

Together, these factors make the coverage prefilter highly robust to hash collisions in practice. While scenarios with extremely dense occupancy may increase collision likelihood, such cases did not arise in our benchmarks, and our empirical sweep demonstrates that the method maintains stable performance across a wide range of hash sizes.

| Hash size | PSNR | SSIM | LPIPS |
|---|---|---|---|
| $2^{11}$ | 21.555 | 0.617 | 0.446 |
| $2^{13}$ | 21.593 | 0.618 | 0.446 |
| $2^{15}$ | 21.507 | 0.616 | 0.447 |
| $2^{17}$ | 21.261 | 0.612 | 0.450 |
| $2^{19}$ | 21.282 | 0.610 | 0.451 |
| No Collision | 21.325 | 0.610 | 0.450 |

Table S4: **Performance comparison across different hash feature sizes**.

$\pi^3$ **implementation.** To evaluate how the choice of dense correspondence model affects our pipeline, we replaced MASt3R with $\pi^3$ (Wang et al., 2025a), a recent VGGT-based matcher that predicts poses, point maps, and tracks directly from image pairs. Unlike triangulation-based MASt3R, $\pi^3$ provides dense 3D predictions without requiring multi-view geometry, but its outputs lie in an internal coordinate frame and thus require alignment to the global COLMAP frame via ICP.

A full re-evaluation using $\pi^3$ shows that reconstruction quality remains highly similar to the MASt3R-based variant, with differences typically within ±0.1 across PSNR, SSIM, and LPIPS (see Table below). Interestingly, the two backbones introduce complementary error characteristics: MASt3R benefits from accurate scale and pairwise consistency inherited from COLMAP poses, whereas $\pi^3$ produces stable background predictions but is sensitive to alignment drift. These error modes counterbalance each other, resulting in comparable end-to-end performance.

Because coverage is measured over sparse, occupied voxels, the advantage of $\pi^3$'s denser predictions in background regions is not fully reflected in the final metrics. Overall, this ablation indicates that while the correspondence backbone affects intermediate behavior, it does not strongly influence the final reconstruction quality within our current setup. This observation suggests several promising extensions, including improved global alignment (e.g., SLAM-style refinement), joint optimization of camera poses and SA-Points, or lighter correspondence models for improved efficiency.

| Backbone | PSNR $\uparrow$ | SSIM $\uparrow$ | LPIPS $\downarrow$ |
|---|---|---|---|
| MASt3R (Ours) | 21.325 | 0.610 | 0.450 |
| Pi$^3$ | 21.179 | 0.609 | 0.451 |

Table S5: **Ablation on correspondence backbone.** Comparison between the MASt3R-based pipeline and the $\pi^3$-based variant. Results show differences within $\pm 0.1$ across PSNR, SSIM, and LPIPS.

**Statistics of Active view selection for each dataset.** We provide per-scene statistics for Active view selection on Mip-NeRF 360 dataset, NeRF Synthetic, and Extended dataset in Table S6 S7& S8. Each experiment is repeated four times with different random seeds.

# F SUPPLEMENTARY VIDEO OVERVIEW

To complement the static visualizations provided in this document, we include a supplementary video that offers dynamic and comprehensive renderings of our results. This video is intended to provide a deeper visual understanding of the improvements achieved by our proposed SA-ResGS framework across various evaluation scenarios.

The video includes:

- **Novel View Rendering.** We present extended 360-degree novel view trajectories, captured along spiral and circular camera paths, which go beyond the discrete test views shown in the main paper

| Methods | PSNR | | | | | | | | |
|---|---|---|---|---|---|---|---|---|---|
| | bicycle | bonsai | counter | flowers | garden | kitchen | room | stump | treehill |
| random | $18.642 \pm 0.295$ | $21.086 \pm 1.547$ | $20.544 \pm 0.450$ | $15.916 \pm 0.583$ | $21.131 \pm 0.242$ | $21.908 \pm 0.671$ | $22.256 \pm 0.575$ | $20.027 \pm 0.841$ | $18.211 \pm 0.401$ |
| ACP | $19.290 \pm 0.213$ | $22.158 \pm 0.101$ | $21.390 \pm 0.382$ | $16.543 \pm 0.192$ | $21.542 \pm 0.105$ | $21.279 \pm 0.464$ | $21.868 \pm 0.604$ | $20.575 \pm 0.339$ | $18.278 \pm 0.455$ |
| MUSIQ | $18.332 \pm 0.146$ | $20.088 \pm 0.375$ | $20.452 \pm 0.190$ | $16.946 \pm 0.069$ | $20.736 \pm 0.245$ | $22.804 \pm 0.194$ | $22.057 \pm 0.097$ | $19.217 \pm 0.297$ | $18.016 \pm 0.269$ |
| CrossScore | $18.485 \pm 0.933$ | $24.021 \pm 0.330$ | $22.480 \pm 0.124$ | $17.222 \pm 0.221$ | $21.876 \pm 0.099$ | $22.994 \pm 0.117$ | $22.924 \pm 0.254$ | $21.003 \pm 0.544$ | $18.681 \pm 0.596$ |
| FisherRF | $18.715 \pm 0.153$ | $23.125 \pm 0.733$ | $21.613 \pm 0.110$ | $16.616 \pm 0.158$ | $21.459 \pm 0.101$ | $23.123 \pm 0.361$ | $22.500 \pm 0.642$ | $20.230 \pm 0.316$ | $18.396 \pm 0.209$ |
| Ours | $18.182 \pm 0.116$ | $24.564 \pm 0.025$ | $22.742 \pm 0.029$ | $16.930 \pm 0.154$ | $22.182 \pm 0.035$ | $24.182 \pm 0.111$ | $24.513 \pm 0.110$ | $20.605 \pm 0.224$ | $18.789 \pm 0.374$ |

(a) Average PSNR on Mip-NeRF 360 (4 trials per scene)

| Methods | SSIM | | | | | | | | |
|---|---|---|---|---|---|---|---|---|---|
| | bicycle | bonsai | counter | flowers | garden | kitchen | room | stump | treehill |
| random | $0.412 \pm 0.008$ | $0.758 \pm 0.037$ | $0.719 \pm 0.011$ | $0.318 \pm 0.012$ | $0.578 \pm 0.008$ | $0.774 \pm 0.022$ | $0.782 \pm 0.017$ | $0.457 \pm 0.028$ | $0.457 \pm 0.009$ |
| ACP | $0.429 \pm 0.006$ | $0.791 \pm 0.011$ | $0.746 \pm 0.007$ | $0.334 \pm 0.005$ | $0.596 \pm 0.003$ | $0.757 \pm 0.010$ | $0.779 \pm 0.015$ | $0.476 \pm 0.012$ | $0.458 \pm 0.008$ |
| MUSIQ | $0.400 \pm 0.003$ | $0.726 \pm 0.016$ | $0.723 \pm 0.006$ | $0.333 \pm 0.002$ | $0.549 \pm 0.009$ | $0.782 \pm 0.009$ | $0.780 \pm 0.004$ | $0.423 \pm 0.011$ | $0.459 \pm 0.008$ |
| CrossScore | $0.397 \pm 0.034$ | $0.836 \pm 0.002$ | $0.772 \pm 0.003$ | $0.340 \pm 0.006$ | $0.593 \pm 0.006$ | $0.803 \pm 0.004$ | $0.811 \pm 0.004$ | $0.494 \pm 0.021$ | $0.465 \pm 0.010$ |
| FisherRF | $0.411 \pm 0.004$ | $0.810 \pm 0.016$ | $0.751 \pm 0.003$ | $0.331 \pm 0.005$ | $0.573 \pm 0.004$ | $0.790 \pm 0.007$ | $0.773 \pm 0.017$ | $0.461 \pm 0.015$ | $0.457 \pm 0.006$ |
| Ours | $0.396 \pm 0.003$ | $0.841 \pm 0.003$ | $0.783 \pm 0.001$ | $0.334 \pm 0.003$ | $0.584 \pm 0.002$ | $0.822 \pm 0.002$ | $0.825 \pm 0.003$ | $0.473 \pm 0.008$ | $0.457 \pm 0.004$ |

(b) Average SSIM on Mip-NeRF 360

| Methods | LPIPS | | | | | | | | |
|---|---|---|---|---|---|---|---|---|---|
| | bicycle | bonsai | counter | flowers | garden | kitchen | room | stump | treehill |
| random | $0.566 \pm 0.001$ | $0.364 \pm 0.024$ | $0.382 \pm 0.010$ | $0.612 \pm 0.011$ | $0.417 \pm 0.002$ | $0.288 \pm 0.018$ | $0.357 \pm 0.009$ | $0.556 \pm 0.016$ | $0.561 \pm 0.006$ |
| ACP | $0.557 \pm 0.003$ | $0.343 \pm 0.008$ | $0.362 \pm 0.008$ | $0.598 \pm 0.003$ | $0.410 \pm 0.003$ | $0.303 \pm 0.008$ | $0.358 \pm 0.012$ | $0.548 \pm 0.006$ | $0.563 \pm 0.008$ |
| MUSIQ | $0.579 \pm 0.002$ | $0.394 \pm 0.014$ | $0.384 \pm 0.004$ | $0.601 \pm 0.002$ | $0.429 \pm 0.004$ | $0.288 \pm 0.007$ | $0.381 \pm 0.007$ | $0.573 \pm 0.005$ | $0.567 \pm 0.005$ |
| CrossScore | $0.605 \pm 0.045$ | $0.318 \pm 0.003$ | $0.355 \pm 0.003$ | $0.609 \pm 0.002$ | $0.418 \pm 0.006$ | $0.269 \pm 0.002$ | $0.354 \pm 0.003$ | $0.541 \pm 0.010$ | $0.564 \pm 0.007$ |
| FisherRF | $0.571 \pm 0.002$ | $0.332 \pm 0.013$ | $0.356 \pm 0.002$ | $0.603 \pm 0.003$ | $0.420 \pm 0.002$ | $0.278 \pm 0.005$ | $0.370 \pm 0.008$ | $0.556 \pm 0.009$ | $0.562 \pm 0.003$ |
| Ours | $0.594 \pm 0.002$ | $0.313 \pm 0.002$ | $0.342 \pm 0.001$ | $0.618 \pm 0.002$ | $0.431 \pm 0.002$ | $0.277 \pm 0.043$ | $0.338 \pm 0.002$ | $0.570 \pm 0.004$ | $0.574 \pm 0.003$ |

(c) Average LPIPS on Mip-NeRF 360

Table S6: **Scene-wise quantitative results on Mip-NeRF 360**. Each subtable reports PSNR, SSIM, and LPIPS respectively. Values are averaged over four trials per scene.

and supplementary figures. These renderings highlight the effectiveness of our physically grounded view selection and residual supervision in preserving structural consistency and photometric quality across challenging viewpoints.

- **Ablation Study Comparisons.** To illustrate the impact of each component, we show side-by-side comparisons of different model variants under continuous camera movement. These scenes demonstrate the robustness and fidelity improvements from residual supervision and surface-aware physically grounded view selection, especially under sparse-view or occluded regions.

## USE OF LARGE LANGUAGE MODELS

All substantive content was authored by the researchers, with the Large Language Models employed solely for auxiliary polishing of sentence-level expressions for better readability.

| Methods | PSNR | | | | | | |
|---|---|---|---|---|---|---|---|
| | chair | drums | hotdog | lego | materials | mic | ship |
| random | $24.626 \pm 3.489$ | $20.918 \pm 1.347$ | $30.397 \pm 1.161$ | $28.812 \pm 0.285$ | $20.013 \pm 0.646$ | $23.813 \pm 1.236$ | $25.348 \pm 0.830$ |
| ACP | $25.989 \pm 0.219$ | $19.176 \pm 0.604$ | $24.409 \pm 0.755$ | $23.474 \pm 0.104$ | $19.125 \pm 0.295$ | $22.687 \pm 0.476$ | $24.168 \pm 0.726$ |
| MUSIQ | $27.899 \pm 0.164$ | $20.948 \pm 1.108$ | $30.308 \pm 0.140$ | $28.796 \pm 0.159$ | $20.268 \pm 0.433$ | $23.219 \pm 0.437$ | $25.219 \pm 0.335$ |
| CrossScore | $24.521 \pm 0.531$ | $18.817 \pm 0.094$ | $28.799 \pm 0.661$ | $27.629 \pm 0.985$ | $18.326 \pm 0.187$ | $23.011 \pm 0.378$ | $25.120 \pm 0.325$ |
| FisherRF | $27.066 \pm 0.981$ | $21.844 \pm 0.608$ | $30.998 \pm 0.371$ | $26.108 \pm 2.142$ | $20.516 \pm 0.488$ | $24.153 \pm 0.668$ | $25.645 \pm 0.408$ |
| Ours | $28.303 \pm 0.280$ | $22.948 \pm 0.304$ | $31.195 \pm 0.166$ | $29.703 \pm 0.450$ | $21.206 \pm 0.187$ | $26.267 \pm 0.271$ | $26.437 \pm 0.042$ |

(a) Average PSNR on NeRF Synthetic dataset

| Methods | PSNR | | | | | | |
|---|---|---|---|---|---|---|---|
| | chair | drums | hotdog | lego | materials | mic | ship |
| random | $0.932 \pm 0.019$ | $0.878 \pm 0.018$ | $0.962 \pm 0.005$ | $0.937 \pm 0.001$ | $0.814 \pm 0.014$ | $0.896 \pm 0.015$ | $0.829 \pm 0.014$ |
| ACP | $0.920 \pm 0.001$ | $0.849 \pm 0.012$ | $0.920 \pm 0.007$ | $0.850 \pm 0.002$ | $0.804 \pm 0.010$ | $0.860 \pm 0.005$ | $0.785 \pm 0.011$ |
| MUSIQ | $0.940 \pm 0.003$ | $0.880 \pm 0.010$ | $0.961 \pm 0.001$ | $0.938 \pm 0.002$ | $0.817 \pm 0.006$ | $0.867 \pm 0.003$ | $0.820 \pm 0.013$ |
| CrossScore | $0.930 \pm 0.003$ | $0.826 \pm 0.008$ | $0.951 \pm 0.003$ | $0.925 \pm 0.004$ | $0.776 \pm 0.007$ | $0.848 \pm 0.011$ | $0.823 \pm 0.007$ |
| FisherRF | $0.945 \pm 0.003$ | $0.885 \pm 0.012$ | $0.965 \pm 0.002$ | $0.908 \pm 0.021$ | $0.810 \pm 0.005$ | $0.898 \pm 0.007$ | $0.834 \pm 0.007$ |
| Ours | $0.949 \pm 0.002$ | $0.903 \pm 0.004$ | $0.965 \pm 0.001$ | $0.941 \pm 0.004$ | $0.826 \pm 0.010$ | $0.918 \pm 0.002$ | $0.846 \pm 0.002$ |

(b) Average SSIM on NeRF Synthetic dataset

| Methods | PSNR | | | | | | |
|---|---|---|---|---|---|---|---|
| | chair | drums | hotdog | lego | materials | mic | ship |
| random | $0.070 \pm 0.016$ | $0.111 \pm 0.009$ | $0.068 \pm 0.004$ | $0.076 \pm 0.001$ | $0.181 \pm 0.008$ | $0.108 \pm 0.008$ | $0.202 \pm 0.007$ |
| ACP | $0.075 \pm 0.002$ | $0.125 \pm 0.006$ | $0.109 \pm 0.003$ | $0.121 \pm 0.002$ | $0.198 \pm 0.007$ | $0.126 \pm 0.002$ | $0.213 \pm 0.006$ |
| MUSIQ | $0.062 \pm 0.002$ | $0.109 \pm 0.007$ | $0.071 \pm 0.002$ | $0.077 \pm 0.002$ | $0.190 \pm 0.005$ | $0.119 \pm 0.003$ | $0.204 \pm 0.003$ |
| CrossScore | $0.069 \pm 0.002$ | $0.125 \pm 0.003$ | $0.088 \pm 0.004$ | $0.086 \pm 0.003$ | $0.213 \pm 0.005$ | $0.123 \pm 0.003$ | $0.203 \pm 0.004$ |
| FisherRF | $0.059 \pm 0.003$ | $0.108 \pm 0.006$ | $0.066 \pm 0.003$ | $0.093 \pm 0.011$ | $0.177 \pm 0.008$ | $0.111 \pm 0.003$ | $0.199 \pm 0.003$ |
| Ours | $0.058 \pm 0.001$ | $0.099 \pm 0.002$ | $0.066 \pm 0.001$ | $0.078 \pm 0.002$ | $0.175 \pm 0.002$ | $0.095 \pm 0.002$ | $0.199 \pm 0.001$ |

(c) Average LPIPS on NeRF Synthetic dataset

Table S7: **Scene-wise quantitative results on NeRF Synthetic dataset**. Each subtable reports PSNR, SSIM, and LPIPS respectively. Values are averaged over four trials per scene.

| Methods | PSNR | | | | | | |
|---|---|---|---|---|---|---|---|
| | ballroom | barn | francis | horse | playroom | ponche | truck |
| random | $16.970 \pm 0.320$ | $17.639 \pm 0.592$ | $18.474 \pm 0.361$ | $19.119 \pm 0.513$ | $18.506 \pm 0.311$ | $19.981 \pm 0.899$ | $21.737 \pm 0.451$ |
| ACP | $17.485 \pm 0.107$ | $18.804 \pm 0.389$ | $18.977 \pm 0.551$ | $20.251 \pm 0.151$ | $20.807 \pm 0.574$ | $21.578 \pm 0.113$ | $19.325 \pm 0.137$ |
| MUSIQ | $16.443 \pm 0.235$ | $18.021 \pm 0.694$ | $18.271 \pm 0.708$ | $18.693 \pm 0.206$ | $19.131 \pm 0.162$ | $18.438 \pm 0.394$ | $20.793 \pm 0.169$ |
| CrossScore | $18.099 \pm 0.085$ | $19.708 \pm 0.253$ | $18.546 \pm 0.151$ | $20.354 \pm 0.119$ | $19.842 \pm 0.150$ | $19.899 \pm 0.809$ | $21.517 \pm 0.525$ |
| FisherRF | $17.250 \pm 0.326$ | $19.208 \pm 0.748$ | $18.708 \pm 0.139$ | $19.834 \pm 0.396$ | $19.334 \pm 0.179$ | $19.833 \pm 0.828$ | $22.017 \pm 0.096$ |
| Ours | $18.281 \pm 0.179$ | $18.611 \pm 0.963$ | $19.803 \pm 0.489$ | $20.015 \pm 0.232$ | $20.689 \pm 1.105$ | $20.906 \pm 1.067$ | $22.115 \pm 0.256$ |

(a) Average PSNR on Extended datasets

| Methods | PSNR | | | | | | |
|---|---|---|---|---|---|---|---|
| | ballroom | barn | francis | horse | playroom | ponche | truck |
| random | $0.551 \pm 0.012$ | $0.613 \pm 0.013$ | $0.747 \pm 0.007$ | $0.756 \pm 0.011$ | $0.679 \pm 0.012$ | $0.753 \pm 0.018$ | $0.757 \pm 0.008$ |
| ACP | $0.563 \pm 0.003$ | $0.632 \pm 0.009$ | $0.762 \pm 0.014$ | $0.792 \pm 0.006$ | $0.770 \pm 0.008$ | $0.752 \pm 0.004$ | $0.705 \pm 0.008$ |
| MUSIQ | $0.524 \pm 0.008$ | $0.624 \pm 0.013$ | $0.744 \pm 0.013$ | $0.749 \pm 0.008$ | $0.700 \pm 0.005$ | $0.723 \pm 0.004$ | $0.741 \pm 0.002$ |
| CrossScore | $0.605 \pm 0.004$ | $0.669 \pm 0.006$ | $0.774 \pm 0.002$ | $0.792 \pm 0.003$ | $0.724 \pm 0.004$ | $0.753 \pm 0.012$ | $0.759 \pm 0.005$ |
| FisherRF | $0.555 \pm 0.013$ | $0.653 \pm 0.017$ | $0.760 \pm 0.005$ | $0.777 \pm 0.007$ | $0.709 \pm 0.006$ | $0.777 \pm 0.007$ | $0.709 \pm 0.006$ |
| Ours | $0.614 \pm 0.007$ | $0.629 \pm 0.018$ | $0.768 \pm 0.007$ | $0.779 \pm 0.007$ | $0.741 \pm 0.016$ | $0.765 \pm 0.013$ | $0.761 \pm 0.403$ |

(b) Average SSIM on Extended datasets

| Methods | PSNR | | | | | | |
|---|---|---|---|---|---|---|---|
| | ballroom | barn | francis | horse | playroom | ponche | truck |
| random | $0.385 \pm 0.007$ | $0.448 \pm 0.013$ | $0.410 \pm 0.006$ | $0.286 \pm 0.009$ | $0.348 \pm 0.011$ | $0.466 \pm 0.019$ | $0.388 \pm 0.006$ |
| ACP | $0.376 \pm 0.002$ | $0.430 \pm 0.006$ | $0.402 \pm 0.010$ | $0.256 \pm 0.005$ | $0.448 \pm 0.006$ | $0.397 \pm 0.003$ | $0.329 \pm 0.008$ |
| MUSIQ | $0.411 \pm 0.006$ | $0.450 \pm 0.012$ | $0.417 \pm 0.011$ | $0.298 \pm 0.008$ | $0.338 \pm 0.003$ | $0.506 \pm 0.004$ | $0.401 \pm 0.001$ |
| CrossScore | $0.352 \pm 0.003$ | $0.400 \pm 0.006$ | $0.383 \pm 0.005$ | $0.257 \pm 0.002$ | $0.318 \pm 0.003$ | $0.466 \pm 0.014$ | $0.386 \pm 0.003$ |
| FisherRF | $0.389 \pm 0.009$ | $0.416 \pm 0.018$ | $0.399 \pm 0.005$ | $0.273 \pm 0.007$ | $0.324 \pm 0.004$ | $0.479 \pm 0.016$ | $0.387 \pm 0.001$ |
| Ours | $0.350 \pm 0.005$ | $0.436 \pm 0.018$ | $0.394 \pm 0.008$ | $0.277 \pm 0.015$ | $0.338 \pm 0.041$ | $0.437 \pm 0.031$ | $0.002 \pm 0.002$ |

(c) Average LPIPS on Extended datasets

Table S8: **Scene-wise quantitative results on Extended datasets**. Each subtable reports PSNR, SSIM, and LPIPS respectively. Values are averaged over four trials per scene.

