# OpenReview forum: "SA-ResGS: Self-Augmented Residual 3D Gaussian Splatting for Next Best View Selection"
_ICLR.cc/2026/Conference — ICLR 2026 Conference Desk Rejected Submission_

### Official Review · Reviewer_2qKx · 2025-10-29

**Soundness:** 2
**Presentation:** 2
**Contribution:** 2
**Rating:** 2
**Confidence:** 5

**Summary:**

This paper introduces SA-ResGS, a method for active scene reconstruction with 3D Gaussian Splatting that combines a geometry-aware prefilter for next-best-view (NBV) selection with an uncertainty-guided residual supervision strategy during training. The prefiltering stage builds “self-augmented points” by establishing correspondences between a trained camera view and a lightly extrapolated render; these points populate a voxelized, hash-encoded coverage space that favors candidate views offering genuinely new surface coverage before any uncertainty scoring is applied. The learning component renders both a full image and a subset image that prioritizes Gaussians considered uncertain (based on opacity/scale), which is intended to inject stronger gradients into weakly supervised splats without modifying the renderer. Experiments on Mip-NeRF 360 and a mixed set from Deep Blending / Tanks & Temples report small but sometimes consistent gains on PSNR/SSIM, with less consistent improvements on LPIPS. A runtime table suggests that candidate selection becomes faster due to the prefilter, although the per-iteration training cost increases because of the residual supervision procedure.

**Strengths:**

The paper’s main strength is the attempt to stabilize early NBV decisions by introducing a physically grounded coverage prefilter that does not depend on potentially unreliable uncertainty estimates. This is a clean and reusable idea that could be integrated into other active-reconstruction pipelines. The residual supervision mechanism is also practical, as it works at the level of data selection without modifying the renderer, and it is easy to plug into existing 3DGS training code. Finally, the candidate selection phase appears to benefit from the prefilter, which would be a useful engineering contribution.

**Weaknesses:**

A major weakness of this work is that several core design choices are heuristic and insufficiently analyzed:

1. The coverage prefilter relies on a hash-encoded voxel grid that can suffer from collisions and is sensitive to voxel size, yet no robustness study regarding the parameter is provided.

2. The uncertainty-guided residual supervision selects Gaussians using opacity/scale; however, the author did not discuss the superiority against Fisher-infomation based uncertainty. [2,3]

3. The pipeline uses MASt3R for dense correspondences without justifying this choice against strong alternatives. For example, VGGT [1] provides dense, geometry-aware matches/tracks and might be more stable under larger baselines or yield denser, less noisy SA-Points in low-texture areas; conversely, lighter matchers such as LoFTR/LightGlue [4] could reduce runtime and memory at some quality cost. Without a head-to-head comparison, it’s unclear whether the reported gains stem from SA-ResGS itself or from the particular strengths/quirks of MASt3R. In short, the generalizability of the method is undermined by not showing results with at least one alternative dense correspondence backbone.

4. Most ablations are confined to Mip-NeRF 360; there is little evidence that the component-level gains carry over to Deep Blending or Tanks & Temples under the same protocol. The selection interval (the number of iterations of adding a view) is not varied or studied, even though it can materially affect both quality and runtime. In coverage estimation, the voxel size and thresholds are used without ablation, raising concerns about portability beyond the evaluated settings.

The experimental scope is not well aligned with common practice in active NeRF/NBV evaluation. The paper focuses primarily on Mip-NeRF 360, whereas much prior work, such as ActiveNeRF and FisherRF, treats NeRF-Synthetic as a primary benchmark specifically to enable comparisons across methods. Without results on NeRF-Synthetic under the standard protocol (dataset split, selection cadence, and training schedule), it is difficult to assess whether the reported gains carry over to the object-centric dataset. The paper should include NeRF-Synthetic results with the same NBV schedule and strong baselines.

The uncertainty-calibration experiments are weakly supported. The AUSE analysis is shown on a single scene and does not include detailed uncertainty–error correlation plots across multiple datasets.

The quantitative improvements are modest and inconsistent across metrics, raising questions about the practical impact of the method. While PSNR/SSIM sometimes tick up, LPIPS is not consistently improved and even degrades on some scenes. The paper also reports results with a single fixed random seed and does not provide standard deviations or confidence intervals, so it is hard to know whether the observed deltas are statistically meaningful or within run-to-run noise.

The qualitative evidence is similarly underwhelming. The supplementary visualization video shows results that are incremental and often visually hard to distinguish from the baseline; for example, on the Bonsai scene the differences are subtle at best. The paper does not provide region-wise quantitative metrics for these highlighted areas (e.g., masked PSNR/LPIPS on the zoom boxes), which would make the visual comparisons more diagnostic.

The runtime analysis is split and, therefore, unconvincing. The candidate-selection phase appears faster due to prefiltering, but per-iteration training becomes more expensive with residual supervision. Without an end-to-end latency comparison for a complete active reconstruction pipeline on the same hardware, with the same number of added views, selection interval, and total iterations, the practical benefit remains unproven.

[1] Wang, Jianyuan, et al. "Vggt: Visual geometry grounded transformer." CVPR. 2025.

[2] Jiang, Wen, Boshu Lei, and Kostas Daniilidis. "Fisherrf: Active view selection and mapping with radiance fields using fisher information." ECCV, 2024.

[3] Li, Ruiqi, and Yiu-ming Cheung. "Variational multi-scale representation for estimating uncertainty in 3d gaussian splatting." NeurIPS 2024.

[4] Sun, Jiaming, et al. "LoFTR: Detector-free local feature matching with transformers." CVPR. 2021.

**Questions:**

1. Could you report results on the standard benchmark of active radiance field reconstruction, the NeRF-Synthetic dataset?

2. How sensitive is performance to the coverage-grid parameters (voxel size, dilation radius) and hashing choices? A small robustness study would help demonstrate that the gains are not brittle.

3. Why did you choose opacity/scale as the uncertainty proxy for the residual subset? Did you evaluate alternatives such as Fisher-information based uncertainty, and if so, how do they compare?

4. The method relies on MASt3R correspondences with extrapolated renders, especially early in training. Could you justify this design choice over methods like VGGT and LoFTR?

---

> ### Author Response · Authors · 2025-11-23
> **Author response (1/7)**
>
> We thank reviewer 2qKx for acknowledging the strengths of our physically grounded coverage prefilter and the practical design of the residual supervision mechanism. We appreciate the reviewer’s positive feedback on the plug-and-play nature of our approach and its usefulness for candidate selection in active reconstruction pipelines. Below, we address the concerns raised by reviewer 2qKx.
>
> > ### **(W1) Robustness Analysis of the Hash-Encoded Voxel Grid**
>
> We thank the reviewer for raising this important point regarding the robustness of the hash-encoded voxel grid used in the coverage prefilter. To evaluate sensitivity to hash-grid collisions, we conducted additional experiments on the Mip-NeRF 360 dataset by varying the hash size from $2^{11}$ to $2^{19}$, as well as testing a no-collision configuration.
>
> | Hash size | PSNR   | SSIM  | LPIPS |
> |-----------|--------|-------|-------|
> | $2^{11}$ | 21.555 | 0.617 | 0.446 |
> | $2^{13}$ | 21.593 | 0.618 | 0.446 |
> | $2^{15}$ | 21.507 | 0.616 | 0.447 |
> | $2^{17}$ | 21.261 | 0.612 | 0.450 |
> | $2^{19}$ | 21.282 | 0.610 | 0.451 |
> | **No Collision** | 21.325 | 0.610 | 0.450 |
>
> Across this range, which represents a over 200× difference in hash capacity, the reconstruction quality and coverage estimation remained largely stable, with no consistent degradation pattern. We found that the effect of hash collisions is smaller than expected due to a structural property of our pipeline:
> 1. we convert only the occupied voxels into hashed indices, skipping the vast majority of empty voxels. Since the occupancy ratio varies significantly across scenes, the effective load on the hash table is typically low.
> 2. Also, we utilizes Fisher-based fine selection part, which further reduces residual errors. These results collectively show that the coverage prefilter is robust to hash collisions and that the method is not sensitive to the chosen hash size in practice. We include this experiment and discussion in the revised version.
>
> That said, we agree with the reviewer that in scenes with substantially higher occupancy density, hash collisions may become more influential. Although this scenario did not occur in our benchmarks, and our empirical sweep shows robustness even under large variations in hash size, we have added the analysis in Supplementary **`Section E` (Ablation on Hash Encoding Size)**. We appreciate the reviewer’s suggestion, which helped strengthen our analysis of the hash-grid parameter.
>
> In addition to hash size, the reviewer also mentions the broader set of parameters involved in constructing the voxel grid. Among these, the dilation parameter is fixed to 2 across all scenes in our current implementation. However, we agree that the voxel-grid resolution indeed depends on the scene scale. As also noted in our limitation section, this component is inherently parameter-dependent, and the expressiveness of the grid representation—hence the final performance—can vary with voxel resolution. This challenge is not unique to our approach and similarly appears in other grid-based representations such as Instant-NGP and TensoRF. We acknowledge this as a limitation in our method as well, as stated in the revised manuscript.

---

> ### Author Response · Authors · 2025-11-23
> **Author response (2/7)**
>
> > ### **(W2) Ablation on ResGS**
>
> We thank the reviewer for highlighting this point. Fisher-based uncertainty can be powerful guidance for view selection, but it is not suitable for the per-iteration residual supervision required in ResGS for the following reasons.
>
> **1. FisherRF is computationally infeasible for residual learning**
>
> Fisher information requires a full render and backpropagation over all candidate views. Meanwhile, 3DGS continuously clones, prunes, and updates Gaussians, which means that Fisher information would need to be recomputed at every iteration to remain valid. As shown in Table 5 of the revised manuscript, even periodic Fisher evaluation is expensive. Performing this at each iteration would make training prohibitively slow. In contrast, residual supervision must run every iteration, which makes Fisher impractical for this purpose.
>
> **2. Fisher uncertainty is unstable early in training**
>
> In the early stages of 3DGS training: geometry is sparse, opacity is uncalibrated, and visibility changes abruptly due to continuous Gaussian cloning and pruning. As demonstrated in the view selection results in Figure S1-S4, FisherRF view selection itself is not stable due to dynamics of 3DGS.
>
> In contrast, we define the uncertainty of a Gaussian using two physically grounded attributes—opacity and scale—which directly determine visibility, rendering contribution, and gradient strength in 3DGS. These attributes provide per-iteration, per-Gaussian signals that remain stable throughout training, making them suitable and practically superior for residual supervision.

---

> ### Author Response · Authors · 2025-11-23
> **Author response (3/7)**
>
> > ### **(W3) Use recent correspondence models?**
>
> We thank the reviewer for the constructive suggestion. We agree that several recent dense correspondence models, such as VGGT and lighter matchers like LoFTR and LightGlue, are strong alternatives to MASt3R. Evaluating different backbones is helpful for understanding the generality and limitations of our pipeline.
>
> Our focus in this work is the SA-ResGS framework, which can in principle operate with a variety of dense matchers. Motivated by the reviewer’s comment, we replaced MASt3R with $\pi^{3}$ [1], an improved VGGT-based model, and re-evaluated the entire pipeline. $\pi^{3}$ predicts camera poses, point maps, and tracks directly from image pairs, which allows us to estimate 3D coverage without triangulation. However, because these predictions lie in the model’s internal coordinate system, an additional alignment step (via a sparse COLMAP point cloud and ICP) is required.
>
> The comparison between the MASt3R-based and $\pi^{3}$-based variants shows that the final reconstruction quality is very similar (as shown in the table below):
>
> | **Backbone**     | **PSNR** ↑ | **SSIM** ↑ | **LPIPS** ↓ |
> |------------------|------------|------------|--------------|
> | MASt3R (Ours)    | 21.325     | 0.610      | 0.450        |
> | $\pi^{3}$       | 21.179     | 0.609      | 0.451        |
>
> with differences typically within ±0.1 across PSNR, SSIM, and LPIPS. Rather than indicating that the two backbones behave identically, we found that each introduces different sources of error:
>
> - MASt3R provides more accurate absolute scale and camera-pair consistency because triangulation uses ground-truth COLMAP camera poses, although triangulation can amplify errors in background regions or for large-baseline view pairs.
>
> - $\pi^{3}$, in contrast, produces dense and stable 3D predictions in background regions, but the required alignment to the global COLMAP frame introduces its own source of drift or mismatch.
>
> These distinct error modes appear to counterbalance each other in practice, resulting in comparable performance across scenes. Additionally, because coverage is computed over sparse, occupied voxels, the advantage of $\pi^{3}$’s denser background predictions is not fully reflected in the final metrics, further contributing to the observed similarity.
>
> Overall, these findings indicate that the choice of correspondence backbone affects intermediate behavior in different ways, but does not significantly change end-to-end performance within our current setup. At the same time, this experiment highlights promising future directions, such as integrating more accurate alignment methods (e.g., SLAM-style refinement), exploring joint optimization of camera pose and SA-Points, or using lighter correspondence models for computational efficiency, as suggested by the reviewer. We believe these represent promising avenues to further extend SA-ResGS beyond the current formulation. We update the analysis above in the **`Appendix E`** and **`Table S5`**
>
> [1] Wang et al., $\pi^3$: Scalable Permutation-Equivariant Visual Geometry Learning, Arxiv 2025

---

> ### Author Response · Authors · 2025-11-23
> **Author response (4/7)**
>
> > ### **(W4) Ablation on additional datasets**
>
> We thank the reviewer for raising this point. We agree that demonstrating component-level robustness beyond Mip-NeRF 360 is important. To address this, we conducted the same ablations on the extended benchmarks, including Deep Blending and Tanks & Temples, and we observed the same qualitative and quantitative trends across all components, as summarized in **`Appendix D`** (**`Table S1`**). These datasets contain more diverse camera trajectories and a wider range of scene scales than Mip-NeRF 360, making them a stronger test bed for evaluating the generality of our method.
>
> | **Methods**               | Sec. 3.2 | Sec. 3.3 | **PSNR** | **SSIM** | **LPIPS** |
> |---------------------------|:-------:|:-------:|---------:|---------:|----------:|
> | FisherRF$^{\dagger}$     |    -    |    -    | 19.293   | 0.715    | 0.378     |
> | $^{\ddagger}$+ResGS     |    -    |    ✓    | 19.462   | 0.719    | 0.381     |
> | $^{\dagger}$+ResGS      |    -    |    ✓    | 19.887   | 0.730    | 0.374     |
> | $^{\dagger}$+SA-HashGS  |    ✓    |    -    | 19.890   | 0.730    | 0.364     |
> | $^{\dagger}$+SA-ResGS   |    ✓    |    ✓    | **20.305** | **0.740** | **0.362** |
>
> ( We note that the Barn scene was excluded from the results below because some frequently failed for reconstruction during ablation runs.)
>
> Consistent with the trends reported in the main paper (**`Sec. 4.3`**), the extended ablations confirm three key observations:
> - **Residual learning consistently improves performance under identical view sequences.**
>
>  Comparing FisherRF$^{†}$ and $^{‡}$+ResGS (first two rows), we observe that adding ResGS yields clear performance gains, reinforcing that the residual module provides stable benefits independent of the selection strategy.
>
> - **Applying only one component leads to suboptimal improvements due to complementary roles.**
>
>  When ResGS or SA-HashGS is applied in isolation (third and fourth rows), improvements become inconsistent. This behavior reflects the known sensitivity of each component to view selection quality and reconstruction difficulty: ResGS alone cannot fully correct errors when early view selections are unstable, while SA-HashGS improves coverage but does not resolve under-supervision in ambiguous regions.
>
> - **The combination of SA-HashGS and ResGS delivers the strongest and most stable improvements.**
>
>  As shown in the final row, jointly applying both components consistently achieves the best performance across scenes. This confirms that physically grounded selection and uncertainty-aware residual supervision address complementary failure modes and work synergistically.
>
> We have incorporated these extended ablation results and explanations into the revised manuscript (**`Appendix D`**, **`Table S1`**). We appreciate the reviewer’s suggestion, as it helped us strengthen the completeness and clarity of our analysis.

---

> ### Author Response · Authors · 2025-11-23
> **Author response (5/7)**
>
> > ### **(W5) Experiments on NeRF Synthetic dataset**
>
> > The experimental scope ... The paper should include NeRF-Synthetic results with the same NBV schedule and strong baselines.
>
> We thank the reviewer for raising this important point regarding evaluation on NeRF-Synthetic, which is commonly used in prior active NeRF and NBV studies such as FisherRF. While NeRF-Synthetic is a widely adopted benchmark, it is also highly constrained: the camera distribution is fixed, viewpoints are densely sampled around a single object, and the scenes are fully controlled and object-centric. These characteristics make it less representative of the real scenarios where next-best-view selection is practically applied. For this reason, we initially focused on more challenging and unconstrained datasets such as Mip-NeRF 360, Deep Blending, and Tanks & Temples.
>
> Nevertheless, we acknowledge the importance of evaluating on established benchmarks. In response to the reviewer’s request, we conducted new experiments on NeRF-Synthetic datasets. To ensure consistency across datasets, we generated sparse point clouds for all NeRF-Synthetic scenes using COLMAP and trained our method on top of the resulting reconstructions. (We note that Ficus fails to reconstruct reliably under COLMAP, and therefore we excluded it and evaluated on the remaining scenes.)
>
> Following the standard FisherRF configuration, we start from 4 initial views and select 1 view every 100 epochs until reaching 20 total views in total. Under this protocol, we compared our method against strong NBV baselines, including ACP, MUSIQ, CrossScore, and FisherRF.
> The averaged results across scenes are shown below:
>
> | **Category** | **Methods** | **PSNR ↑** | **SSIM ↑** | **LPIPS ↓** |
> |--------------|-------------|-----------:|-----------:|------------:|
> | Rule-based   | Random      | 24.530     | 0.890      | 0.118       |
> |   | ACP         | 22.886     | 0.856      | 0.139       |
> | 2D-based     | MUSIQ       | 25.137     | 0.889      | 0.117       |
> |     | CrossScore  | 24.293     | 0.882      | 0.126       |
> | 3D-based     | FisherRF    | 25.166     | 0.888      | 0.118       |
> |     | Ours        | **26.496** | **0.907**  | **0.110**   |
>
> Consistent with our findings on Mip-NeRF 360, Deep Blending, and Tanks & Temples, our method achieves the best performance across all metrics on NeRF-Synthetic. These results confirm that the improvements provided by SA-ResGS generalize well beyond the 360° dataset and hold under the standard object-centric NBV protocol.
>
> We note that there is a differences in the input setting, while fair. Our evaluation uses a COLMAP-based reconstruction pipeline, which introduces realistic reconstruction noise, such as incomplete or uneven initial point density, structural holes in low-texture regions, and small pose drift. FisherRF relies heavily on the quality of the initial 3D geometry, so its performance is also affected by the COLMAP-derived point cloud. This explains why FisherRF achieves higher scores in the official Blender benchmark but slightly lower performance under our unified COLMAP-driven protocol. Importantly, our method is substantially more robust to imperfect initial geometry, which is also observed consistently across all datasets.
>
> We added the evaluation protocol to the revised manuscript **`Appendix A.2.1`** to avoid confusion and included the results in **`Table 1(b)`**. We thank the reviewer again for the helpful suggestion.

---

> ### Author Response · Authors · 2025-11-23
> **Author response (6/7)**
>
> > ### **(W7-1) Low LPIPS**
>
> > The quantitative improvements ... While PSNR/SSIM sometimes tick up, LPIPS is not consistently improved and even degrades on some scenes.
>
> We thank the reviewer for this thoughtful observation. While SA-ResGS shows strong improvements over the baselines, we acknowledge that the gains in perceptual metrics such as SSIM and LPIPS do not always reach the absolute state-of-the-art across all scenes.
>
> This behavior is related to the characteristics of our design. As stated in **`Sec 4.1`- Result part (`line number 410-412`)**, SA-ResGS focuses on improving structural consistency through physically grounded view selection and residual supervision. These components often lead to smoother and more stable geometry, which benefits PSNR and sometimes SSIM. However, the same smoothing effect can slightly reduce high-frequency texture details, which LPIPS is particularly sensitive to. Similar trade-offs have been reported in previous 3DGS work, where improving geometric stability can modestly influence perceptual metrics.
>
> We also agree that tuning certain elements of the pipeline, such as residual weighting, uncertainty thresholds, or the strength of geometric filtering, may help narrow the gap in SSIM and LPIPS. In preliminary tests, adjusting these parameters does shift the balance between texture sharpness and geometric smoothness, although this often comes with reduced stability or consistency. We plan to explore these directions in future work, as they represent promising avenues for enhancing perceptual quality without compromising stability.
> We appreciate the reviewer’s suggestion and will clarify this trade-off and its underlying cause in the revised manuscript.
>
> > ### **(Q3-2) Repeated experiments**
>
> > The paper also reports results with a single fixed random seed and does not provide standard deviations or confidence intervals ....
>
> We thank the reviewer for emphasizing the importance of reporting robustness across multiple runs. In response, we repeated experiments on all dataset, including the Mip-NeRF 360, NeRF Synthetic, and Extended dataset (Deep blending and Tank and Temples dataset), four times using different random seeds and updated the results accordingly.
>
> The trends we originally reported remain consistent, and notably, our method also shows lower standard deviation compared to the baselines, indicating improved stability. We have added per-scene statistics to the **appendix D, `Table S6-S8`**, and **`Table 2`** has been updated to report the mean over four runs.
>
> We appreciate the reviewer for encouraging a more rigorous statistical evaluation.
>
> > ### **(W8) Qualitative results are incremental**
>
> > The qualitative evidence ... make the visual comparisons more diagnostic.
>
> We thank the reviewer for the valuable feedback regarding the qualitative evaluation. Since our work tackles both active view selection and 3D reconstruction, we aim to demonstrate improvements in both coverage behavior and final reconstruction quality. To make these aspects clearer, we have added the supplementary materials with these visualizations:
>
> In Appendix **`Figure S1–S4`**, we visualize the camera-pose distributions, showing that our method reduces the cluttered or redundant view selections frequently observed in the baselines.
>
> In Appendix **`Figure S5–S9`**, we also provide multiple novel view rendering results to illustrate reconstruction outcomes across the entire capture trajectory.
>
> Regarding the suggestion to report region-wise cropped metrics (for example, masked PSNR or LPIPS on zoomed-in areas), we appreciate the intuition. However, such localized cropping can inadvertently bias comparisons toward specific textures or spatial regions, which may not reflect overall reconstruction quality and may not be fair. To maintain fairness and consistency across scenes, we evaluate all methods using the same full-image metrics for all viewpoints.

---

> ### Author Response · Authors · 2025-11-23
> **Author response (7/7)**
>
> > ### **(W9) Runtime analysis**
>
> > The runtime analysis is split ... Without an end-to-end latency comparison ... remains unproven.
>
> Thank you for pointing out this issue. In the original submission, we reported the runtime of individual components (Table 4), such as view prefiltering and per-iteration training cost, to clarify where the computational differences arise. We agree with the reviewer that an end-to-end latency comparison provides a more convincing assessment of practical efficiency.
>
> Following the reviewer’s suggestion, we conducted a full active-reconstruction runtime measurement on the same hardware, with the same number of added views, same selection interval, and same total training iterations. Specifically, we repeated the experiment five times on the Bonsai scene and report the average end-to-end run time using a single GPU :
> - FisherRF: 31 min 59 sec
> - SA-ResGS (ours): 19 min 45 sec
>
> As discussed in the main paper, this improvement stems from the fact that FisherRF must compute per-Gaussian Fisher information across all candidate views at each selection step, which dominates its runtime. In contrast, our physically grounded prefiltering substantially reduces the number of candidate views that require Fisher evaluation, while residual supervision introduces only a modest per-iteration overhead. The end-to-end measurement confirms that the reduction in candidate-selection cost outweighs the additional training cost, resulting in substantially shorter total reconstruction time.
>
> We also note that our current runtime results are obtained using the default configuration, where residual supervision is applied at every iteration and prefiltering is performed with a relatively conservative threshold. In practice, our framework offers several additional design choices that can further reduce computation, such as applying residual supervision every (k) iterations or adopting a tighter prefiltering ratio to reduce the number of candidate views more aggressively. We did not employ these options in the reported experiments to keep the comparison fair and consistent with the baseline settings, but we believe they provide additional room for improving efficiency without compromising reconstruction quality. We clarified these possible extensions in the Limitation and Future work paragraph.
>
> We have revised the table (**`Table 4` in revised version**) and **`Section 4.4`** with these updates.
>
> > ### **(W6) More experiments on AUSE**
>
> > The uncertainty-calibration experiments ... across multiple datasets.
>
> Thank you for raising this important point. We agree that the current uncertainty-calibration experiment is limited, as it is shown only on a single scene. In the revised manuscript, we have addressed this concern by extending our AUSE evaluation to all nine scenes of the Mip-NeRF 360 dataset, following the same experimental protocol described in Sec. 4.2. These additional results allow us to more comprehensively assess the alignment between predicted uncertainty and actual depth errors under diverse geometric and photometric conditions. Across these scenes, we observe a consistent reduction in AUSE, with averages decreasing from 0.327 (FisherRF) to 0.323 (+ResGS) and 0.297 (+SA-ResGS). This trend demonstrates that the improvements are not scene-specific but generalize across varied geometric structures and lighting conditions.
>
> Furthermore, we also provide additional qualitative comparisons of uncertainty–error correlation across multiple scenes (see Fig. 6), Over multiple scenes, our method produces more spatially aligned uncertainty maps, especially in regions exhibiting high depth error, while the baseline often underestimates uncertainty in these challenging areas. This qualitative patterns further demonstrating consistent improvements when incorporating residual learning (+ResGS) and self-augmented prefiltering (+SA-ResGS). These extended results strengthen our claim that the proposed components systematically enhance uncertainty calibration, rather than improving only on a specific scene.
>
> We believe these additions substantially reinforce the validity of our uncertainty-estimation evaluation.

---

### Official Review · Reviewer_zhib · 2025-10-31

**Soundness:** 2
**Presentation:** 2
**Contribution:** 3
**Rating:** 6
**Confidence:** 3

**Summary:**

This paper introduces SA-ResGS (Self-Augmented Residual 3D Gaussian Splatting), a framework designed to improve uncertainty-aware active mapping and next-best-view (NBV) selection in 3D reconstruction. The authors propose a Self-Augmented Point (SA-Point) mechanism, which triangulates correspondences between a real image and an extrapolated rendered view to produce physically grounded geometry priors for efficient scene-coverage estimation and robust NBV selection. To complement this, they introduce the first residual learning strategy for 3D Gaussian Splatting, which enhances training stability by explicitly reinforcing supervision on uncertain or weakly contributing Gaussians through uncertainty-guided sampling and dual-render residual supervision. Together, these innovations aim to stabilize uncertainty quantification, mitigate vanishing gradients, and improve reconstruction quality under sparse or wide-baseline conditions. Comprehensive experiments and ablations on Mip-NeRF 360, Deep Blending, and Tanks & Temples datasets demonstrate consistent gains in reconstruction fidelity, uncertainty calibration, and computational efficiency over existing NBV and uncertainty-based baselines.

**Strengths:**

Originality:
 The paper makes a notable and original contribution by integrating physical constraints and a residual learning strategy into the active 3D Gaussian Splatting (3DGS) paradigm. The introduction of Self-Augmented Points (SA-Points) for physically grounded next-best-view selection represents a creative and conceptually elegant solution to the long-standing problem of unreliable uncertainty estimation under sparse-view settings. Moreover, the residual supervision mechanism—the first of its kind for 3DGS—extends the notion of residual learning from image-based networks to the 3D splatting domain, addressing vanishing gradient issues in an entirely new context. This combination of geometry-aware and uncertainty-driven reasoning is both novel and insightful, setting a promising new direction for active neural rendering research.

Clarity:
 The paper is well-organized and communicates its ideas clearly. Each component of the proposed framework is described with solid intuition and supported by algorithmic detail, figures, and pseudocode-level explanations that make the method straightforward to reproduce. The authors provide extensive implementation details in both the main text and appendix, including triangulation procedures, voxel hashing, and uncertainty filtering strategies. The logical flow from motivation to methodology and experiments is easy to follow, and the paper succeeds in making a technically complex idea accessible to a broad audience in 3D vision and rendering.

**Weaknesses:**

While the proposed framework is conceptually strong and methodologically sound, the experimental results reveal certain limitations that warrant attention. Specifically, although SA-ResGS achieves clear improvements over several baselines, its quantitative performance does not consistently reach state-of-the-art levels in all metrics—particularly SSIM and LPIPS—suggesting room for further refinement in perceptual quality and structural consistency. Additionally, some experimental descriptions contain minor typographical errors and ambiguous phrasing which I will list in the questions. Addressing these clarity issues would further strengthen the empirical credibility and overall impact of the work.

**Questions:**

1. The paper states that SA-Points mitigate the bias of uncertainty signals caused by incomplete or under-constrained geometry by providing a surface-aware guidance mechanism independent of uncertainty estimation. However, since SA-Points are triangulated using correspondences between a training view and an extrapolated rendered view, wouldn’t their accuracy still depend on the completeness and quality of the underlying geometry? Could the authors clarify how the method ensures stability of SA-Points when the 3DGS geometry is still immature in early training stages?

2. In Table 1(a), the SSIM score of 0.610 is highlighted as the best, though it is not the highest value in the column. Could the authors clarify whether this is a typographical error or if there is a specific reason (e.g., statistical significance, dataset subset, or reporting convention) for highlighting this value?

3. In Table 1(b), the ActiveNeRF baseline is missing, despite being included in Table 1(a). Was this omission intentional due to implementation or dataset compatibility issues, or could the authors provide results for completeness? Including this comparison would strengthen the empirical evaluation.

4. The method does not achieve state-of-the-art results on SSIM and LPIPS metrics, which partially measure perceptual fidelity and structural quality. Could the authors elaborate on the potential causes for this performance gap, and whether tuning the residual weighting parameters or uncertainty thresholds might narrow the difference?

5. The reported result for †+SA-ResGS in Table 2 differs from the corresponding “Ours” entry in Table 1(a), though both appear to represent the full model. Could the authors explain whether these configurations differ or if this discrepancy is due to averaging over different runs or datasets? A clarification would help ensure consistent interpretation of the ablation and main results.

6. Recent NeRF and 3DGS-based active mapping approaches that leverage uncertainty—such as ‘Naruto: Neural Active Reconstruction from Uncertain Target Observations’ and ‘ActiveGAMER: Active Gaussian Mapping through Efficient Rendering’—are not discussed in the related work section. Including a comparison or discussion of these closely related methods would provide valuable context, highlight the distinctions of SA-ResGS, and better situate the proposed framework within the evolving landscape of uncertainty-aware active reconstruction research.

---

> ### Author Response · Authors · 2025-11-23
> **Author response (1/4)**
>
> We thank reviewer zhib for appreciating the originality of integrating physically grounded SA-Points and residual learning into active 3DGS, and for acknowledging the clarity of our methodological presentation. We are encouraged by the reviewer’s recognition of both the conceptual novelty and the reproducibility of our framework. We provide our responses to the reviewer’s productive feedback below.
>
>
> > ### **(W1, Q4) Low SSIM/LPIPS score**
>
> We thank the reviewer for this thoughtful observation. While SA-ResGS shows strong improvements over the baselines, we acknowledge that the gains in perceptual metrics such as SSIM and LPIPS do not always reach the absolute state-of-the-art across all scenes.
>
> This behavior is related to the characteristics of our design. As stated in **`Sec 4.1`- Result part (`line number 410-412`)**, SA-ResGS focuses on improving structural consistency through physically grounded view selection and residual supervision. These components often lead to smoother and more stable geometry, which benefits PSNR and sometimes SSIM. However, the same smoothing effect can slightly reduce high-frequency texture details, which LPIPS is particularly sensitive to. Similar trade-offs have been reported in previous 3DGS work, where improving geometric stability can modestly influence perceptual metrics.
>
> We also agree that tuning certain elements of the pipeline, such as residual weighting, uncertainty thresholds, or the strength of geometric filtering, may help narrow the gap in SSIM and LPIPS. In preliminary tests, adjusting these parameters does shift the balance between texture sharpness and geometric smoothness, although this often comes with reduced stability or consistency. We plan to explore these directions in future work, as they represent promising avenues for enhancing perceptual quality without compromising stability.
>
> We appreciate the reviewer’s suggestion and will clarify this trade-off and its underlying cause in the revised manuscript.

---

> ### Author Response · Authors · 2025-11-23
> **Author response (2/4)**
>
> > ### **(Q1) SA-Points still rely on geometry?**
>
> We appreciate the reviewer’s thoughtful question. We agree that SA-Points inevitably depend on the underlying scene geometry, and their quality can be affected in the very early stages of 3DGS training. Our method does not try to remove this dependency entirely; instead, it is designed to mitigate its impact through several complementary components.
>
> First, SA-Points rely primarily on MASt3R’s dense correspondences rather than on the geometry predicted by 3DGS. MASt3R operates on the ground-truth training image paired with an extrapolated rendered view. Importantly, MASt3R produces a per-correspondence confidence estimate, which can be interpreted as a data-driven prior learned from large-scale training. This confidence prediction reliably down-weights uncertain or noisy matches, enabling SA-Points to remain stable even when early geometry is imperfect.
>
> Second, our pipeline includes a reprojection-error filtering step that discards triangulated points that are not geometrically self-consistent. As shown in **`Sec 4.3`** and **`Table 3`**, we explicitly evaluate robustness under synthetic correspondence noise, and SA-Points remain reliable due to this filtering mechanism.
>
> Third, the proposed residual learning module complements SA-Points by improving geometric stability throughout early training. As highlighted in the ablation studies (**`Table 2, Table S1`**), residual supervision accelerates the refinement of uncertain or weakly constrained Gaussians, indirectly improving the reliability of SA-Points over time.
>
> Taken together, these components allow SA-Points to function robustly despite imperfect early geometry. We will revise the paper to clarify this dependency and to better communicate the empirical evidence supporting SA-Points’ stability. As stated in Limitation part in manuscript, we also agree that exploring stronger correspondence models or additional geometric regularizers would be a valuable extension.
>
> > ### **(Q3) Why not active NeRF on additional dataset?**
>
> We thank the reviewer for raising this concern. We initially attempted to reproduce the ActiveNeRF results using the publicly available implementation included in the FisherRF codebase. However, we encountered repeated execution issues during our attempts, and despite several efforts—including modifying the environment and testing multiple configurations—we were unable to obtain a fully functioning setup on the extended datasets.
>
> For this reason, we reported only the previously published ActiveNeRF numbers from FisherRF in **`Table 1(a)`**. We agree that presenting this baseline inconsistently across tables may cause confusion. To avoid this, we have removed the ActiveNeRF rows from the main tables in the revised manuscript and have clarified this point accordingly.
>
> We appreciate the reviewer for highlighting this issue and helping us improve the clarity and consistency of our empirical evaluation.

---

> ### Author Response · Authors · 2025-11-23
> **Author response (3/4)**
>
> > ### **(Q5) Difference between Table 1(a) and Table 2**
>
> We thank the reviewer for carefully pointing out this inconsistency. After re-checking the results, we found that the value reported for $^{†+}$SA-ResGS in **`Table 2`** was computed with one scene (treehill) unintentionally omitted from the 9-scene evaluation set.  We have corrected this oversight and updated the numbers in the revised version using the complete set.
>
> Importantly, the corrected result remains fully consistent with our observations. The overall trend and conclusions are unchanged, and the performance differences discussed in the paper still hold. We appreciate the reviewer for helping us improve the clarity and consistency of the reported results.
>
> In addition, in response to **reviewer 2qKx’s** request for stronger statistical reliability, we performed repeated experiments and updated **`Table 1-(a)`** using the mean values across multiple runs instead of a single trial. This revision replaces the previously reported single-run numbers and provides a more stable and statistically representative comparison. We note that this update does not affect the qualitative trends or the conclusions of the paper; all components maintain the same relative performance differences as originally described.

---

> ### Author Response · Authors · 2025-11-23
> **Author response (4/4)**
>
> > ### **(Q6) Additional related works**
>
> We sincerely thank the reviewer for recommending ActiveGAMER and NARUTO. Both are great works in active reconstruction, and we agree that they are relevant in the broader context of uncertainty-aware or information-driven view selection.
>
> While their problem settings differ from ours in several important ways, for example the use of RGB-D inputs and the focus on active exploration and planning in SLAM-oriented environments, they share the common motivation of prioritizing informative observations under constrained budgets. We believe that their approaches and ours could be complementary, and future extensions may combine their trajectory-level planning strategies with our uncertainty-aware training schedule within a unified framework.
>
> We have added both works to the **`Sec. 2`**, as related works in the revised manuscript. We again thank the reviewer for bringing these valuable references to our attention.
>
> > ### **(Q2) Typography**
>
> Thank you for highlighting these helpful formatting and typographical issues. We have corrected them and updated the revised version accordingly. While addressing this comment, we also carefully reviewed all tables and metrics throughout the paper to ensure that no similar formatting or reporting errors remain. We appreciate the reviewer’s careful and constructive feedback.

---

### Official Review · Reviewer_bGUu · 2025-11-01

**Soundness:** 3
**Presentation:** 3
**Contribution:** 3
**Rating:** 8
**Confidence:** 4

**Summary:**

This paper proposes SA-ResGS, a novel framework for active 3D reconstruction and next-best-view (NBV) selection built upon the 3D Gaussian Splatting (3DGS) paradigm. The key challenge addressed is the instability of uncertainty estimation and sparse geometric coverage during the early training phase of active 3DGS.

The authors introduce two main components:

Self-Augmented Points (SA-Points):
A mechanism that synthetically perturbs the current camera viewpoint to create a virtual stereo pair. Dense correspondences (via MASt3R) between the current and perturbed views are triangulated into pseudo-3D points, which approximate surface coverage. These points are projected to candidate viewpoints, and a hash-based coverage score is computed to identify views that reveal new, unobserved regions.

Residual Supervision for 3DGS (ResGS):
A dual-path rendering supervision scheme in which one full rendering is complemented by a sub-rendering containing randomly sampled and high-uncertainty Gaussians. Both renderings are supervised against the ground truth, effectively amplifying gradients in uncertain or under-fitted regions.

Experiments on Mip-NeRF360, Deep Blending, and Tanks & Temples show consistent PSNR/SSIM improvements over FisherRF and ActiveNeRF, with faster convergence and better uncertainty calibration.

**Strengths:**

Conceptually novel SA-Points mechanism:
The idea of generating pseudo-3D points from a synthetically perturbed virtual view for coverage-aware NBV selection is original and well-motivated. It bridges geometric reasoning and learning-based active view planning in a lightweight manner.

Stability improvement without heavy computation:
The method improves early-stage training stability and uncertainty estimation without adding expensive optimization or extra 3D supervision.

Practical training refinement:
The residual supervision elegantly balances global consistency and local refinement, improving reconstruction detail while maintaining stable gradients.

Comprehensive experiments:
The paper evaluates across multiple benchmarks, ablates both SA-Points and ResGS, and demonstrates performance and efficiency gains over strong baselines.

**Weaknesses:**

ResGS innovation is incremental:
While effective, the residual supervision is conceptually similar to uncertainty-weighted or hard-example reweighting schemes known in NeRF and 3DGS literature; the novelty lies more in integration than in principle.

Lack of robustness and generalization analysis:
No experiments test how SA-Points or ResGS behave when correspondence quality or uncertainty estimation deteriorates.

Ablation depth:
It remains unclear whether the gains are primarily due to SA-Points’ geometric guidance or simply more stable sampling. A finer ablation (e.g., varying perturbation magnitude or replacing MASt3R with a lighter matcher) would help clarify.

**Questions:**

On SA-Points reliability:
How sensitive is the coverage estimation to correspondence errors from MASt3R? Have the authors tested using noisier or lower-quality matchers?

On manual parameters:
What perturbation magnitude is used to synthesize the virtual viewpoint, and how was it chosen? Could this magnitude be adaptive to scene scale or depth range?

On ResGS interpretation:
Can the authors clarify how their residual supervision differs conceptually from existing uncertainty-guided or sample-reweighting methods?

On robustness:
If the dense matching fails (e.g., in dynamic or textureless regions), does the coverage estimation degrade gracefully or bias the NBV selection?

On generalization:
Could SA-ResGS handle unseen domains (e.g., moving scenes, outdoor environments), or is the method tied to static and well-textured indoor data?

Additional related works:
ActiveGAMER: Active gaussian mapping through efficient rendering
NARUTO: Neural active reconstruction from uncertain target observations

---

> ### Author Response · Authors · 2025-11-23
> **Author response (1/4)**
>
> We thank reviewer bGUu for the encouraging evaluation and for highlighting the conceptual novelty of SA-Points, the stability benefits achieved without heavy computation, and the practical effectiveness of our residual supervision. We are grateful for the reviewer’s positive assessment of our comprehensive experiments and overall contributions. We address the concerns raised below and will incorporate these discussions into the revised manuscript.
>
>
> > ### **(W1, Q3) Novelty of the ResGS**
>
> We thank the reviewer for the insightful comment. As mentioned, we agree that the main contribution of ResGS lies in the systematic integration of residual supervision into an active 3DGS pipeline. The residual signal directly influences both Gaussian updates and view-selection decisions, as supported by the ablations in Table 2 and the additional experiments on the extended dataset (**`Table S1`**).
>
> Although ResGS shares high-level goal of emphasizing difficult regions, its mechanism differs fundamentally from uncertainty-based or hard-example reweighting used in NeRF/3DGS. Existing methods only rescale pixel- or sample-level errors, but they cannot overcome a structural issue in 3DGS: the  “Adaptive Density Control”.
>
> In 3DGS, the rule-based pruning process removes Gaussians that receive insufficient gradients, and this occurs once every few hundred iterations. Methods that do not incorporate a skip connection for residual learning ultimately cannot prevent weakly contributing Gaussians from being pruned. Unless these methods explicitly recover or supervise such Gaussians during rasterization, these under-contributing Gaussians inevitably become targets of rule-based opacity pruning which repeats throughout training. Consequently, once gradients are lost, reweighting cannot recover them and unable to avoid repeatedly losing Gaussians through this pruning mechanism.
>
> ResGS instead introduces a dedicated residual supervision pathway. By rendering an auxiliary image in which dominant Gaussians are partially suppressed, we expose under-contributing Gaussians to additional gradients. This provides a distinct correction signal—analogous to residual learning—rather than merely adjusting weights within the original rendering. The residual cues identify persistently under-updated Gaussians, the uncertainty prevents premature pruning, and the view-selection policy reinforces these weak regions with stronger supervision. Furthermore, the residual connection offers a new mechanism for balancing high- and low-frequency signals during optimization—particularly important within the rule-based structure of 3DGS, where Gaussians can otherwise drift toward overly smooth or overly sparse representations.
>
> To repeat, our main contribution is introducing residual supervision to active 3DGS as reviewer bGUu mentioned. But resGS design not only provides a new correction pathway for weak Gaussians but also offers structural advantages by exposing under-contributing Gaussians to persistent gradients, reducing unintended pruning, and strengthening supervision on difficult regions—that existing reweighting-based approaches cannot achieve.

---

> ### Author Response · Authors · 2025-11-23
> **Author response (2/4)**
>
> > ### **(W2-3, Q1, Q4) Generalization analysis (dependency on matching accuracy)**
>
> We thank the reviewer for highlighting the importance of evaluating the robustness of SA-Points and ResGS under degraded correspondence quality.
>
> **(Noise robustness analysis)**
> This analysis is already included in our manuscript, and the results can be directly verified in the simulated robustness experiments presented in **`Sec 4.3`** and **`Table 3`** in the revised manuscripts (which was in Appendix E in original submission). To measure the robustness or dependence on 2D correspondence matching quality, we synthetically add random noise to each matched 2D points, and measure the quality of active view reconstruction in Mip-NeRF 360 dataset. In the noise-injection study shows that adding up to 5 pixels of synthetic noise to MASt3R correspondences does not destabilize reconstruction quality under active view selection. The system degrades smoothly, and the drop begins to observe at noise level of 10 pixels.
>
> **(Design choice to mitigate noise)**
> We fully agree that the pipeline inevitably depends on matching accuracy; however, our method is explicitly designed to mitigate this dependency through three complementary components:
>
> (1) a transformer-based dense matcher with strong global priors,
>
> (2) reprojection-error filtering that removes inconsistent correspondences, and
>
> (3) voxel dilation that preserves spatial support even when many points are filtered out.
>
> These elements collectively contribute to the stability of SA-Points. Even in challenging scenarios such as low-texture or repetitive regions, modern transformer-based matchers (MASt3R, VGGT, CroCo, $\pi^3$) provide significantly more coherent correspondences than traditional keypoint-based methods (i.e., SIFT, SURF), thanks to the vision transformer architecture and large data driven prior. Also, the combination of re-projection based filtering and voxel dilation further prevents next-best-view selection from drifting toward systematic errors, ensuring that moderate correspondence noise does not introduce biased or unstable reconstruction outcomes. Additional experiments on reprojection-error thresholds and their interaction with view selection are also included in **`Appendix E`**.
>
> Overall, our results show that the proposed framework remains stable under realistic levels of correspondence deterioration and degrades gracefully under more severe conditions. The complementary use of dense-matcher robustness, geometric filtering, and voxel-based regularization prevents catastrophic failure and maintains reliable next-best-view selection even when correspondence quality declines.

---

> ### Author Response · Authors · 2025-11-23
> **Author response (3/4)**
>
> > ### **(Q2) Experimental details**
>
> We sincerely thank the reviewer for the constructive suggestion. We described the experimental setup in **`Appendix. Section A.1.1`** of the manuscript. The perturbation magnitude used to synthesize virtual viewpoints was determined empirically and kept fixed across all experiments for consistency. In the camera coordinate system, we apply a translation of 0.25 units along the horizontal and vertical axes, and 0.5 units backward along the z-axis.
>
> We agree that adapting the perturbation magnitude to scene scale or depth range is a meaningful extension. As we noted in a **limitation** in the manuscript, i.e., the manual parameter setup, as the current parameter choice is manually set. We consider this an important future direction, and we believe that a probabilistic or scale-adaptive formulation could provide a principled way to determine perturbation magnitude more flexibly.
>
> > ### **(Q5) Application on Dynamic or Textureless regions**
>
> We thank the reviewer for raising this important point regarding robustness in regions where dense matching may fail.
>
> **(1) Static vs. dynamic scenes.**
>
> Our method is designed for static scene reconstruction, and we acknowledge that handling dynamic scenes is a limitation of our current framework as stated in the Limitation part in manuscripts.
>
> **(2) Textureless or repetitive regions.**
>
> While correspondence failures are a legitimate concern for traditional feature-based methods such as SIFT or SURF, our pipeline relies on modern dense correspondence networks such as MASt3R. These models are built on vision transformers and are significantly more resilient to low-texture surfaces and repetitive structures. By leveraging global patch-level attention, they infer correspondences using contextual information rather than relying solely on distinctive local features. This allows them to maintain stable 2D matching even in regions where classical descriptors fail.
>
> The reliability of such models in generating dense correspondences has made it feasible for us to introduce a novel idea—triangulating Self-Augmented Points (SA-Points) from extrapolated views, which may contain mild distortions. Despite these distortions, transformer-based models are capable of producing sufficiently accurate correspondences to generate well-structured 3D point clouds. As demonstrated in the Playroom scenes (e.g., Deep Blending), the resulting geometry is clean and consistent, validating the effectiveness of our pipeline even in scenes with limited texture or strong repetition.
> Thus, our framework builds on the growing empirical evidence that modern dense correspondence networks, especially those grounded in transformers, are well suited for physically grounded 3D scene understanding, including in the exact cases highlighted by the reviewer.

---

> ### Author Response · Authors · 2025-11-23
> **Author response (4/4)**
>
> > ### **(Q6) Additional related works**
>
> We sincerely thank the reviewer for recommending ActiveGAMER and NARUTO.
> Both are great works in active reconstruction, and we agree that they are relevant in the broader context of uncertainty-aware or information-driven view selection.
>
> While their problem settings differ from ours in several important ways, for example the use of RGB-D inputs and the focus on active exploration and planning in SLAM-oriented environments, they share the common motivation of prioritizing informative observations under constrained budgets. We believe that their approaches and ours could be complementary, and future extensions may combine their trajectory-level planning strategies with our uncertainty-aware training schedule within a unified framework.
>
> We have added both works to the **`Sec. 2`**, as related works in the revised manuscript. We again thank the reviewer for bringing these valuable references to our attention.

---

### Official Review · Reviewer_EVNA · 2025-11-01

**Soundness:** 2
**Presentation:** 3
**Contribution:** 2
**Rating:** 4
**Confidence:** 4

**Summary:**

SA-ResGS introduces a novel framework for next-best-view selection in 3D Gaussian Splatting that addresses uncertainty quantification instability in sparse-view reconstruction. The method generates Self-Augmented Points (SA-Points) via triangulation between training views and extrapolated renders, enabling physically-grounded view selection that reduces dependence on uncertain early-stage model predictions. It proposes the first residual learning strategy for 3DGS, rendering both full-scene and uncertainty-guided Gaussian subsets (90% random + 10% most uncertain) to amplify gradients for weakly-supervised Gaussians, mitigating the vanishing gradient problem.
The two-stage selection pipeline (physical prefiltering via hash-encoded voxel dissimilarity, then uncertainty ranking) decouples view planning from training dynamics. Experiments on active view selection demonstrate that SA-ResGS outperforms state-of-the-art baselines in both reconstruction quality and view selection robustness.

**Strengths:**

1. It clearly points out three specific limitations of existing NBV methods, which is valuable and insightful for readers.

2. The paper introduces an innovative approach to decouple view selection from early-stage uncertainty estimation through Self-Augmented Points (SA-Points). Addresses a real limitation as the early-stage uncertainty estimates in 3DGS are unreliable due to sparse geometry and training instability.

**Weaknesses:**

1. The paper positions itself as contributing to sparse-view 3D reconstruction but only compares against view selection methods, does not compare against specialized sparse-view reconstruction methods for example FSGS, SparseGS, DNGaussian, MVPGaussian, RegGaussian. Need to clarify how much the 'Next Best View Selection' useful, for example, given 20 images, carefully select views (SA-ResGS) + standard training compared with uniform sampling +  strong regularization (FSGS, SparseGS, ...), also need add experiments to carefully select views (SA-ResGS) + all other different sparse-view reconstruction methods to verify that the view selection contribution is orthogonal to regularization methodology.

2. The paper claims to introduce "the first residual supervision framework for 3DGS", but several prior works have proposed to address the vanishing gradient problem in 3DGS, such as pixelSplat, PAPR, DropGaussian as the paper mentioned in related work, but haven't show experiments for 'ResGS' with other methods, also no ablation showing advantage of uncertainty-guided dropout over random dropout (DropGaussian). The "first residual supervision" claim appears overstated. The actual novel contribution—uncertainty-guided structured dropout—is not empirically validated against simpler random dropout baselines.

3. In table2, fixed-order view selection is better than dynamically updated view selection, this is counterintuitive, as dynamic selection should adapt to training progress and outperform fixed sequences. The explanation in paper ‘training improvements alone are insufficient, especially under high computational uncertainty quantification errors’ is very limited, not provides any quantitative analysis, mechanism explanation, or uncertainty visualization to support this claim. And the ablation lacks dynamic_selection + standard_training, making it impossible to determine whether the problem stems from dynamic selection itself, residual supervision interference, or their interaction.

**Questions:**

Please refer to the Weaknesses above.

---

> ### Author Response · Authors · 2025-11-23
> **Author response (1/4)**
>
> We thank reviewer EVNA for recognizing the value of our analysis of the limitations in existing NBV methods and for acknowledging the importance of decoupling view selection from early-stage uncertainty through SA-Points. We appreciate the reviewer’s positive remarks on the insightfulness and relevance of our proposed framework. Below, we address each concern raised by reviewer EVNA.
>
> > ### **(W1) Comparison with sparse-view selection**
>
> We appreciate the reviewer’s insightful suggestion regarding combining SA-ResGS with existing sparse-view reconstruction methods. We agree that such combinations are promising, and we expect that integrating uncertainty-aware view selection with sparse-view priors could further strengthen overall reconstruction under extremely limited inputs.
>
> Our work, however, focuses on a complementary and orthogonal aspect of the problem. Methods such as FSGS, SparseGS, DNGaussian introduce strong priors or modify the optimization objective to regularize highly under-constrained settings. (We tried to find MVPGaussian, and RegGaussian but we couldn't, so if you share the paper reference, we will also look those too). In contrast, SA-ResGS keeps the base 3DGS optimizer unchanged and instead improves (1) the reliability of uncertainty-driven Next-Best-View ranking and (2) the optimization of Gaussians that receive insufficient gradients due to alpha-blending suppression. Our ablations analysis show that physically grounded prefiltering and the residual supervision branch each improve NBV quality independently, and their combination yields compounded benefits—highlighting that these gains arise from improved view distribution, not from additional priors.
>
> We agree that SA-ResGS is, in principle, compatible with sparse-view reconstruction methods. However, a fair combinational study is nontrivial. Sparse-view approaches are highly sensitive to multiview overlap and assume dense or moderately overlapping baselines, whereas SA-ResGS intentionally produces wide-baseline, coverage-maximizing view distributions. Applying existing sparse-view regularizers under these very different geometric conditions leads to unstable behavior unless each method is individually re-tuned and re-validated. Conducting such per-method re-optimization goes beyond the scope of this submission but is essential for obtaining interpretable results, rather than artifacts caused by mismatched assumptions.
>
> We believe the reviewer’s suggestion points to an exciting future direction. Since SA-ResGS introduces no changes to the underlying optimizer, it provides a modular view-selection mechanism that can be integrated with many sparse-view regularization approaches. We plan to explore these combinations in future work, where a dedicated experimental setup can ensure fair tuning and meaningful conclusions.

---

> ### Author Response · Authors · 2025-11-23
> **Author response (2/4)**
>
> > ### **(W2) Novelty of ResGS**
>
> We thank the reviewer for the helpful comment.
>
> First, we clarify that the contribution of ResGS lies in systematically integrating residual supervision into an active 3DGS pipeline, where the residual signal directly influences both Gaussian updates and view-selection. This is supported by the ablations in **`Table 2`** and extended results in **`Table S1`**.
>
> ### **Claim of the first residual supervision framework for 3DGS.**
>
> Prior works such as pixelSplat, PAPR, and DropGaussian do address vanishing gradients. However, none introduces a residual supervision strategy that provides a secondary, explicitly targeted render aimed to route gradients toward Gaussians that would otherwise be suppressed. Our method offers a more principled, objective-driven mechanism to identify under-updated Gaussians within the active training loop. In our pipeline, the full-set render loss ($I_{full}$) is combined with the subset-render loss ($I_{sup}$), jointly guiding the update direction and leading to a distinct operational behavior.
>
> To further clarify this distinction, we included additional comparisons between the our complete supervision ($I_{full} + I_{sup}$) pipeline and a subset-only ablation that uses only subset-only render loss ($I_{sup}$) (see **`Fig. 7`**). This subset-only variant behaves similarly to DropGaussian. Subset-only tends to over-smooths ambiguous regions and loses high-frequency detail. We attribute this to the rule-based characteristics of 3DGS: pruning process remove Gaussians that receive insufficient gradients, and subset-only supervision lacks the feedback mechanisms needed to prevent this, resulting in blurred or uncertain regions.
>
> Specifically, existing methods which do not incorporate a skip connection for residual supervision pathway ultimately cannot prevent weakly contributing Gaussians from being pruned during training. Unless these methods explicitly recover or supervise such Gaussians during the rasterization process, these weakly contributing Gaussians will inevitably become targets of the rule-based opacity pruning innate in 3DGS—an event that occurs roughly once every few hundred iterations. Consequently, methods lacking such a skip connection are structurally unable to avoid repeatedly losing Gaussians through rule-based pruning, unless they are able to supervise those weakly contributing Gaussians sufficiently—and early enough—to recover them before pruning is applied (within the first few hundred iterations), which is not realistically achievable when relying on the random selection behaviours of dropout-like mechanisms in 3DGS.
>
> ### **Empirical validation for the uncertainty-guided structured dropout.**
>
> Regarding uncertainty-guided dropout, we found that the numerical gains over random dropout remain modest. This is not because the two strategies behave similarly, but rather due to the limitations of the evaluation protocol. Each test view contains a mixture of regions that were visible during training and regions that were never observed. Since the latter are fundamentally impossible to reconstruct accurately, their errors dominate the averaged quantitative metrics (e.g., PSNR), masking improvements that occur specifically in the reconstructable, training-visible regions.
>
> In these training-visible regions—where meaningful comparisons can actually be made—the structured dropout strategy maintains noticeably sharper geometry and preserves high-frequency details, whereas random dropout often leads to blurred or floating Gaussians (as shown in **`Fig. S8`** (b) vs. (c)). Thus, the uncertainty-guided dropout still provides a more principled and effective mechanism: it selectively protects geometrically unstable or ambiguous Gaussians that would otherwise remained under optimized.
>
> To reiterate our contribution, the main value of ResGS does not lie in the dropout component alone, but in the synergy that emerges when residual supervision, uncertainty-guided dropout, and active view selection operate jointly. Utilizing both full-set render loss with subset-render loss identify persistently under-updated Gaussians, uncertainty prevents premature pruning, and view selection reinforces such weak regions with stronger supervision. Additionally, integrating residual supervision into the 3DGS optimization process provides a new mechanism for balancing high- and low-frequency signals during optimization—an aspect particularly aligned with the rule-based nature of 3DGS, where Gaussians can otherwise drift toward overly smooth or overly sparse representations.
>
> This integrated design preserves structural details in sparse or challenging regions and mitigates the blurring commonly observed in existing methods, as reflected in our ablations (**`Table 2`**).

---

> ### Author Response · Authors · 2025-11-23
> **Author response (3/4)**
>
> > ### **(W3) Explanation on Ablation study**
>
> Thank you for raising this question regarding the ablation study in **`Table 2`**, particularly why the fixed-order view selection (second row) appears to outperform the dynamically updated view selection (third row). We acknowledge that this result can seem counterintuitive, and that our brief explanation in the paper may have caused confusion. The naming of the variants may also have contributed to misinterpretation. Below, we clarify the intended meaning of each configuration to show that the ablation was designed to isolate individual components of our method rather than compare fixed and dynamic selection as a standalone research question.
>
> **1\) FisherRF$^{†}$ (baseline):** This configuration corresponds to the original FisherRF pipeline. It computes a Fisher-information–based uncertainty score over Gaussians and dynamically selects the next view based on this metric. This row represents dynamic view selection combined with standard 3DGS training, without any residual supervision or prefiltering.
>
> **2\) $^{‡}$+ResGS:** This configuration isolates the effect of our residual learning module (**`Sec. 3.3`**) while keeping the view ordering unchanged. It uses the exact same view sequence produced by FisherRF$^{†}$ (identical to **Method 1**), with all other settings held constant. The performance gain over FisherRF$^{†}$ demonstrates that ResGS provides an independent benefit when evaluated under the same selection conditions.
>
> **3\) $^{†}$+ResGS:** This variant applies our residual learning module (**`Sec. 3.3`**) while continuing to rely on FisherRF$^{†}$ for selecting each next-best view. Our results show that this combination is fragile because next-best-view selection is a sequential decision process. If early selections are incorrect, the resulting errors propagate into all subsequent choices. FisherRF$^{†}$ relies on computational uncertainty, which is particularly unstable during the early training stage when geometry and gradients are still noisy. When unreliable views are selected at the beginning, the resulting coverage gaps and biased sampling accumulate throughout training. In this setting, residual learning alone cannot fully correct the downstream effects, even though ResGS clearly improves reconstruction when the view sequence itself is stable and reliable, as demonstrated by **Method 2**\.
>
> This behavior reinforces our main argument: early-stage reliance on purely computational uncertainty can misguide the view selection trajectory, and once the sequence diverges, later supervision cannot fully recover the lost coverage or correct suboptimal training signals. This motivates the introduction of **Method 4** ($^{†}$+SA-HashGS), which stabilizes early decisions through physically grounded prefiltering before any computational uncertainty scoring is applied.
>
> **4\) $^{†}$+SA-HashGS:** This variant isolates the effect of our prefiltering module (**`Sec. 3.2`**). Instead of relying entirely on Fisher-based computational uncertainty, which is highly vulnerable to noise in the early training stage, we first apply a physically grounded, geometry-aware filtering step to restrict the candidate set to the most promising views. Fisher-based uncertainty is then applied only within this reduced subset. This design mitigates the disadvantages of depending solely on unstable early uncertainty estimates and stabilizes the selection process. Compared to FisherRF$^{†}$, both the prefiltering module (**`Sec. 3.2`**) and the isolated residual learning module (**Method 2, `Sec. 3.3`**) individually contribute a consistent PSNR improvement of approximately 0.21–0.23.
>
> **5\) $^{†}$+SA-ResGS (full model):** This configuration applies both components, SA-HashGS (**`Sec. 3.2`**) and ResGS (**`Sec. 3.3`**). As shown in **`Table 2`**, the two modules work synergistically. The combined model outperforms each individual component by a meaningful margin. This indicates that stable, physically grounded view selection helps prevent early-stage errors caused by unreliable computational uncertainty estimates, and that residual supervision further stabilizes the optimization process. When used together, these two mechanisms reinforce one another and provide stronger performance than when either module is applied alone.
>
> (cont'd)

---

> > ### Author Response · Authors · 2025-11-23
> > **Author response (4/4)**
> >
> > (cont'd)
> >
> > In summary, the ablation study was designed to isolate each proposed component and to examine their individual effects as well as their interactions. The results show that ResGS consistently improves reconstruction quality when evaluated under the same view ordering (row 2 compared with row 1). They also indicate that purely dynamic FisherRF selection becomes unstable even when ResGS is added, despite ResGS being stable and beneficial under a fixed selection order (row 3). In addition, the physically grounded prefiltering module provides a more reliable early selection strategy by avoiding the sensitivity of computational uncertainty scores to early-stage noise (row 4). Finally, combining both components yields the strongest overall performance, confirming that their contributions are complementary (row 5). We agree that clearer terminology would help avoid misunderstandings, and we will revise the manuscript to reflect these explanations more clearly.

---

### Author Response · Authors · 2025-11-23

**We thank the reviewers for their constructive comments and suggestions for improving our work. We appreciate the positive feedback:**
- Clearly Motivation on previous NBV methods (EVNA, zhib)
- Introducing novel algorithm for physically grounded, lightweight stabilizer for early NBV (EVNA, bGUu, zhib, 2qKx)
- Providing plug-and-play residual supervision that improves stability and detail efficiently (bGUu, zhib, 2qKx)
- Clear, well-organized presentation with reproducible details (zhib)
- Thorough experiments and ablations validating performance and efficiency gains (bGUu)

**We have carefully addressed reviewer comments, and we welcome further feedback or clarification. Below, we provide a summary of our initial rebuttal:**
- Additional ablations
  - Extended-dataset ablations → Appendix Sec D, Table S1
  - Hash-encoding analysis → Appendix Sec E, Table S3
  - Backbone variation experiments  → Appendix Sec E, Table S4
- New dataset evaluation - NeRF-Synthetic → Table 1(b), Sec. 4.1; dataset details in Appendix Sec. A.2.1
- End-to-end runtime comparison → Table 4 & Sec. 4.4
- Statistical reliability (multi-run results) → Table 2 note: “four trials for each scene”; statistics in Table S6-S8
- Related Works & corrections → Section 2 (Related Works), corrected tables throughout main text

**We think that our submission has become more convincing thanks to the invaluable feedback. We again thank the reviewers for their time and comments.**

---

> ### Author Response · Authors · 2025-12-03
> **Summary of Our Rebuttal for Area Chair**
>
> Dear Area Chair,
>
> We offer this summary to support the assessment of our manuscript, the original reviews, and our comprehensive responses.
> To contextualize our contributions and the revisions we made, we first outline the key challenges inherent to active view selection in 3DGS reconstruction and how our method is designed to directly address them.
>
> ***Challenge 1 : Unreliable uncertainty estimation***
>
> - In the early-stage training phase, when scene coverage is incomplete, uncertainty estimation becomes biased and unstable, hindering effective view planning.
> - **Our Solution : Physically grounded view selection using Self-Augmented Points (SA-Points).** Utilize explicit geometric coverage cues, enabling physically plausible and coverage-aware NBV decision
> - **Acknowledgement :** *“Clear motivation.”* (EVNA, zhib) : *“A novel approach for NBV.”* (EVNA, bGUu, zhib, 2qKx)
>
> ***Challenge 2: Under-utilized supervision of weakly contributing Gaussians***
>
> - Existing NBV pipelines rarely convert uncertainty signals into actionable supervision, leaving weakly contributing Gaussians consistently under-optimized.
> - **Our Solution :  Residual learning for 3DGS.** We propose the first targeted residual supervision mechanism for 3D Gaussian Splatting, explicitly reinforcing weak Gaussians and stabilizing fine-detail recovery.
> - **Acknowledgement :** *“An efficient plug-and-play residual supervision module for fine-detail reconstruction.”* (bGUu, zhib, 2qKx)
>
> The initial reviews raised valid concerns. We have undertaken a substantial revision to address these points. Below, we summarize the most significant updates:
>
> > ### **Generalizability (bGUu, zhib, 2qKx)**
>
> * *Initial Concerns:* Instability may stem from (i) Correspondence error (ii) Different view distribution, and (iii) randomness of 3DGS
> * **Clarification and Revision**:
>   * Clarify that original manuscript already conducted noise robustness experiments →  **`Sec 4.3`** and **`Table 3`**
>   * Conduct new dataset evaluation \- NeRF-Synthetic → **`Table 1(b)`**, **`Sec. 4.1`**; dataset details in **`Appendix Sec. A.2.1`**
>   * Show statistical reliability (multi-run results) → **`Table 1(a)`** note: “four trials for each scene”; statistics in **`Table S6`**
> * **Impact :**
>   * Demonstrating the robustness to geometric, correspondence error thanks to the systemic design of the SA-ResGS
>   * Demonstrate more stable and generalizable performance among various scenarios
>
> > ### **Clarification on Novelty of ResGS (EVNA, bGUu)**
>
> * *Initial Concerns* : Novelty of the ResGS
> * **Clarification :**
>   * First integration of residual supervision into an active 3DGS pipeline.
>   * Residual supervision pathway tailored for 3DGS learning dynamics.
> * **Revision :** Add empirical validation for the uncertainty-guided structured dropout and residual supervision → **`Fig 7`**, **`Fig S8`**
> * **Impact :** These demonstrate that our work provides novel attributes to Next Best View selection tasks
>
> > ### **Clarification on Computational Burden (2qKx)**
>
> * *Initial Concerns* : Need to provide an end-to-end active view selection pipeline runtime.
> * **Clarification and Revision** : Compute end-to-end runtime comparison (**`Table 4`** & **`Sec. 4.4`**)
> * **Impact :** Demonstrate SA-ResGS achieves geometry grounded active reconstruction while reducing computational burden
>
> > ### **Additional Ablation study for justification for design choice (2qKx)**
>
> * *Initial Concerns* : Need ablation study on design choices for (a) Hash-encoding size, (b) backbone variation
> * **Clarification and Revision :**
>   * Compute Hash-encoding analysis → **`Appendix Sec. E`**, **`Table S3`**
>   * Compute Backbone variation experiments → **`Appendix Sec. E`**, **`Table S4`**
> * **Impact :**
>   * Showing the robustness in change of design choice
>   * Demonstrate the versatility to the correspondence model opening the future possibilities
>
> > ### **About reconstruction quality (zhib, 2qKx)**
>
> * *Initial Concerns* : Show relatively lower SSIM/LPIPS in some scenes, and clarify of quality improvement
> * **Clarification and Revision** :
>   * Reduction is marginal, Structural consistency is prioritized in ResGS → noted in **`Sec. 4.1 (lines 410–412)`**.
>   * Active view selection require both coverage and reconstruction quality → reflected in **`Fig. S1–S7`**.
> * **Impact :**
>   * Clearly articulate the strengths and limitations of the ResGS approach.
>   * Highlight our method’s advantages in the active view selection.
>
> All additional comments regarding experimental details (bGUu), typography (zhib), and other minor issues have been fully addressed in the revision.
> We believe that these revisions thoroughly address the reviewers’ concerns and significantly strengthen the clarity, rigor, and empirical support of the manuscript.
> Thank you again for your time and consideration.
>
> Best regards,
>
> Authors

---

### Note · Program_Chairs · 2026-01-17
**Submission Desk Rejected by Program Chairs**

The following references in this submission do not refer to real documents and/or have major errors in bibliographic information:

 Zihang Wang et al. Uncertainty-aware ensemble nerf for active view planning. In IEEE Conf. Comput. Vis. Pattern Recog., 2023.